# Prostaglandin signaling regulates nephron segment patterning of renal progenitors during zebrafish kidney development

Shahram Jevin Poureetezadi[1,2], Christina N Cheng[1,2], Joseph M Chambers[1,2], Bridgette E Drummond[1,2], Rebecca A Wingert[1,2]*

[1]Department of Biological Sciences, University of Notre Dame, Notre Dame, United States; [2]Center for Stem Cells and Regenerative Medicine, Center for Zebrafish Research, University of Notre Dame, Notre Dame, United States

**Abstract** Kidney formation involves patterning events that induce renal progenitors to form nephrons with an intricate composition of multiple segments. Here, we performed a chemical genetic screen using zebrafish and discovered that prostaglandins, lipid mediators involved in many physiological functions, influenced pronephros segmentation. Modulating levels of prostaglandin E2 ($PGE_2$) or $PGB_2$ restricted distal segment formation and expanded a proximal segment lineage. Perturbation of prostaglandin synthesis by manipulating Cox1 or Cox2 activity altered distal segment formation and was rescued by exogenous $PGE_2$. Disruption of the $PGE_2$ receptors Ptger2a and Ptger4a similarly affected the distal segments. Further, changes in Cox activity or $PGE_2$ levels affected expression of the transcription factors *irx3b* and *sim1a* that mitigate pronephros segment patterning. These findings show for the first time that $PGE_2$ is a regulator of nephron formation in the zebrafish embryonic kidney, thus revealing that prostaglandin signaling may have implications for renal birth defects and other diseases.

*For correspondence: rwingert@nd.edu

**Competing interests:** The authors declare that no competing interests exist.

## Introduction

The kidney serves central functions in metabolic waste excretion, osmoregulation, and electrolyte homeostasis. Vertebrate kidney organogenesis is a dynamic process involving the generation of up to three distinct structures that originate from the intermediate mesoderm (IM) (*Saxen, 1987*). In mammals, a pronephros, mesonephros, and metanephros develop in succession. Of these structures, the pronephros and mesonephros both eventually disintegrate, leaving the metanephros as the adult kidney. In contrast, lower vertebrates such as fish and amphibians only form a pronephros and mesonephros, which are active during embryogenesis and larval stages, respectively, and the mesonephros subsequently serves as the adult organ (*Dressler, 2006*).

During the progression of vertebrate kidney ontogeny, composition of the basic renal functional unit, termed the nephron, remains largely similar across species (*Desgrange and Cereghini, 2015*). Nephrons contain a renal corpuscle that filters the blood, a tubule that modifies the filtrate solution, and a collecting duct (*Romagnani et al., 2013*). The tubule portion of the nephron is configured along its proximo-distal axis with specific groupings of cells, termed segments, which possess unique physiological roles in solute reabsorption and secretion. While the organization of proximal and distal nephron segments is broadly conserved (*Romagnani et al., 2013*), the genetic and molecular mechanisms that regulate formation of each segment lineage have yet to be fully described (*Costantini and Kopan, 2010*).

The zebrafish embryonic pronephros is a useful model to delineate the processes that regulate vertebrate nephron segmentation because it is anatomically simple, being comprised of only two

nephrons (*Gerlach and Wingert, 2013*). Further, the transparent nature of zebrafish embryos, their *ex utero* development, and the ease at which large numbers can be obtained and managed, are all features that readily facilitate renal development and disease studies (*Pickart and Klee, 2014*; *Poureetezadi and Wingert, 2016*). The zebrafish pronephric tubule has four discrete tubule segments: a proximal convoluted tubule (PCT), proximal straight tubule (PST), distal early (DE), and distal late (DL) (*Wingert et al., 2007*) (*Figure 1A*). The proximal segments are homologous to the PCT and PST in mammals, while the distal segments are homologous to the mammalian thick ascending limb (TAL) and distal convoluted tubule (DCT), respectively (*Wingert et al., 2007*; *Wingert and Davidson, 2008*).

During zebrafish kidney development, renal progenitors arise rapidly from the IM and undergo a mesenchymal to epithelial transition (MET) to engender the tubule by 24 hr post fertilization (hpf) (*McKee et al., 2014*; *Gerlach and Wingert, 2014*). Prior to this, the renal progenitors undergo complex segment lineage patterning events, beginning with their segregation into rostral and caudal subdomains, a process that is orchestrated by the morphogen retinoic acid (RA) which is locally secreted by the adjacent paraxial mesoderm (PM) (*Wingert et al., 2007*; *Wingert and Davidson, 2011*). Modulating levels of RA affects the specification of renal progenitors, inducing proximal segment lineage formation over distal, which can be accentuated by the addition of exogenous all-trans RA, while distal fates are induced over proximal by inhibiting endogenous production of RA through the application of the biosynthesis inhibitor N,N-diethlyaminobenzaldehyde (DEAB) (*Wingert et al., 2007*; *Wingert and Davidson, 2011*). Through expression profiling and subsequent functional studies, several transcription factors have been mapped as acting downstream of RA signaling to regulate pronephros segmentation and epithelial fate choice, including *hepatocyte nuclear factor-1 beta* (paralogues *hnf1ba* and *hnf1bb*), *iroquois homeobox 3b* (*irx3b*), *mds1/evi1 complex* (*mecom*), *single minded family bHLH transcription factor 1a* (*sim1a*), and *t-box 2* (paralogues *tbx2a* and *tbx2b*), among others (*Wingert and Davidson, 2011*; *Naylor et al., 2013*; *Li et al., 2014*; *Kroeger and Wingert, 2014*; *Cheng and Wingert, 2015*; *Marra and Wingert, 2016*; *Marra et al., 2016*; *Drummond et al., 2017*). Despite these advances, the identity of the other essential signals that control renal progenitor fate decisions has remained elusive (*Cheng et al., 2015*).

Historically, prostaglandins have been defined as functionally diverse molecules that regulate an array of biological tasks, including inflammation and vasoregulation (*Funk, 2001*; *Tootle, 2013*). With regard to the adult kidney, prostaglandins regulate many aspects of renal physiology, ranging from tubular transport processes to hemodynamics (*Nasrallah et al., 2007*). Prostaglandins are lipid mediators produced by the sequential actions of a series of enzymes, and exert their effects by paracrine or autocrine signaling through distinct G-protein coupled receptors (*Funk, 2001*; *Tootle, 2013*). More specifically, there are five major prostaglandins produced from the precursor arachidonic acid (AA) by the enzymes Prostaglandin-endoperoxide synthase one or Prostaglandin-endoperoxide synthase 2a (Ptgs1 and Ptgs2a in zebrafish, also known as cyclooxygenases COX-1 and COX-2 in mammals) followed by subsequent processing by particular synthases (*Funk, 2001*; *Tootle, 2013*). Each bioactive prostanoid interacts with one or more G-protein coupled membrane receptors (*Funk, 2001*; *Tootle, 2013*). For example, COX activity on AA can generate the intermediate $PGH_2$, from which the $PGE_2$ bioactive can be produced by the prostaglandin E synthase (Ptges) (*Figure 1E*). $PGE_2$ will signal by subsequent interactions with Ptger G-protein coupled receptors including EP1, EP2, EP3 and EP4 (known as Ptger1-4 in zebrafish) on receiving cells (*Figure 1E*) (*Funk, 2001*; *Tootle, 2013*; *Yang et al., 2013*).

While prostaglandin biosynthesis and signal transduction have been extensively studied in both healthy and diseased adult tissues (*Matsuoka and Narumiya, 2007*; *Smyth et al., 2009*), knowledge of their roles in development have been more challenging to ascertain for several reasons. Firstly, although it is thought that various factors that produce prostaglandins are broadly expressed during ontogeny, precise knowledge about the spatiotemporal progression of particular pathway components is incomplete. Secondly, there is a substantial void in our understanding due to the results of murine loss of function studies where genetic disruptions of components within the prostaglandin pathway was associated with observably normal development. This led to the hypothesis that maternal prostaglandin sources had rescued embryogenesis, thereby complicating the use of mammalian models to study prostaglandin requirements during ontogeny. The importance of reevaluating prostaglandin signaling in kidney formation has been emphasized by a recent report that COX-2 dosage

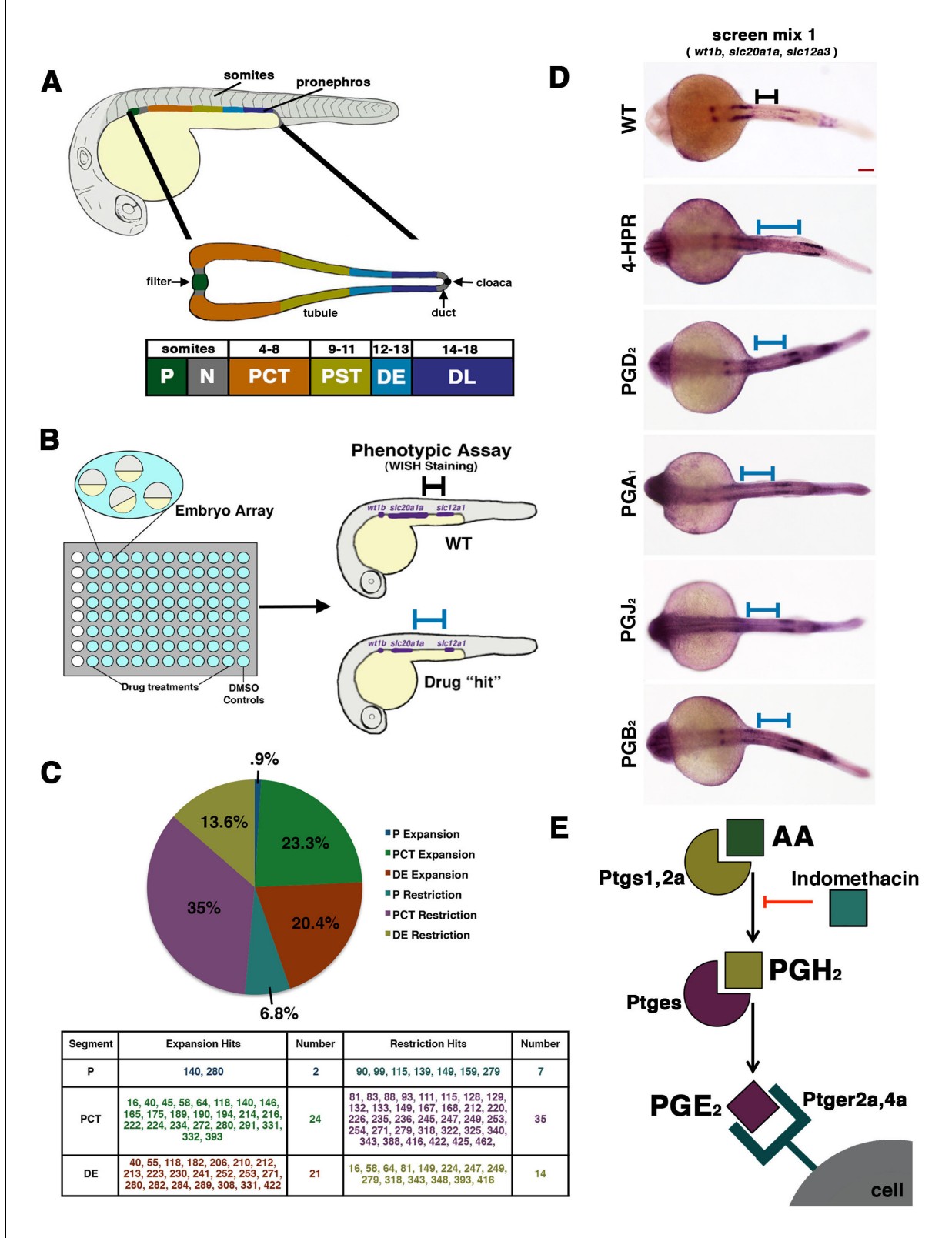

**Figure 1.** A novel small molecule screen reveals that prostaglandins alter nephron patterning. (**A**) A diagram detailing the segmentation of the pronephros in relation to somites within the zebrafish embryo. Arrows indicate the blood filter, duct, and cloaca. (**B**) A schematic of the chemical genetic screen used for evaluating small molecules. Embryos were arrayed in 96-well plates and then exposed to drugs diluted in E3 medium from 60% epiboly to 24 hpf, where the embryos were then fixed and underwent WISH using a riboprobe cocktail to detect the P (*wt1b*), PCT (*slc20a1a*), and DE

*Figure 1 continued*

(*slc12a1*). Black and blue bars are used to illustrate changes between the WT embryo and an embryo with a patterning phenotype, respectively. (C) A pie graph and table denoting the number and percentage of small molecules hits from the chemical screen that expanded or restricted the P (blue and teal), PCT (green and purple) or DE (red and yellow). (D) WISH in 24 hpf stage embryos to detect the P (*wt1b*), PCT (*slc20a1a*), and DE (*slc12a1*) in WTs and those treated with 4-HPR, $PGD_2$, $PGA_1$, $PGJ_2$, and $PGB_2$. A black or blue bar was used to notate the segment change between the WT and drug treated embryos, respectively. Red scale bar, 70 μm. (E) Schematic showing example components of prostaglandin production and signaling. The precursor arachidonic acid (AA) interacts with either the Ptgs1 or Ptgs2a enzyme to generate an intermediate moiety, with the example here being $PGH_2$. The intermediate interacts with a subsequent enzyme to produce the bioactive prostanoid molecule. Here, we depict the prostaglandin E synthase, Ptges, creating the bioactive prostaglandin $PGE_2$ that can transduce signals through binding several G-protein coupled receptors such as Ptger2a and Ptger4a. Other receptors work with other bioactive prostaglandins. Indomethacin is a nonselective Cox (Ptgs1/Ptgs2a) inhibitor that prevents prostaglandin biosynthesis.

The following source data is available for figure 1:

**Source data 1.** Compilation of chemical screen phenotypic data.

is critical for murine metanephros development, though it is presently enigmatic whether there are requirement(s) for discrete stages of nephrogenesis (*Slattery et al., 2016*).

In lieu of the challenges of using mammalian systems to delineate the roles of prostaglandin signaling during development, considerable insights in vertebrates have nevertheless been achieved recently through research using the zebrafish model. Most notably, there have been transformative revelations regarding the conserved roles of prostaglandin signaling during definitive blood formation, where $PGE_2$ was found to regulate hematopoietic stem cell (HSC) development and function (*North et al., 2007*). A chemical genetic screen in zebrafish also identified the prostaglandin pathway as a modifier of endoderm organogenesis, where in subsequent work it was found that $PGE_2$ activity controls opposing cell fate decisions in the developing pancreas and liver through the *ep4a* receptor (also known as *ptger4a*), which derive from a bipotential endoderm progenitor (*Garnaas et al., 2012*; *Nissim et al., 2014*). Other than these studies, there is little known about how prostaglandin signaling may affect cell fate decisions during the emergence of other vertebrate tissues.

Here, we report the discovery that $PGE_2$ signaling has potent effects in regulating proximal and distal segment formation during nephrogenesis in the developing zebrafish kidney. Using the zebrafish embryo for gain and loss of function studies, in addition to whole mount *in situ* hybridization (WISH) to profile gene expression, we uncovered that the Cox enzymes Ptgs1 and Ptgs2a, as well as the $PGE_2$ receptors Ptger2a and Ptger4a, are necessary to properly establish distal nephron segment boundaries during pronephros genesis. Further, we found that addition of $PGE_2$ was sufficient to rescue distal segmentation in Ptgs1 and Ptgs2a deficient zebrafish. Interestingly, treatment with exogenous $PGE_2$ or $PGB_2$ during nephrogenesis induced a striking expansion of a proximal tubule segment lineage in a dosage-dependent manner. Taken together, this work reveals for the first time that alterations in $PGE_2$ signaling, and possibly other prostaglandins as well, has important consequences for the developing nephron.

## Results

### Chemical genetic screen reveals that prostaglandin levels affect nephron development

To date, much remains unknown concerning the factors that control nephron segment development and cell fate decisions. The zebrafish pronephros is an experimentally tractable system to interrogate the genetic factors that regulate nephrogenesis because of its simple, conserved tubule structure, with two proximal segments and two distal segments (*Figure 1A*) (*Ebarasi et al., 2011*; *Drummond and Wingert, 2016*). The nephrons share a blood filter comprised of podocyte cells (P), followed by a neck (N) segment that transports fluid into the tubule, and finally a pronephric duct (PD) that drains caudally at the cloaca (C), a common exit for the kidney and gut in the embryo (*Figure 1A*, middle panel). Nephron segment fates are established by the 24 hpf stage, based on the expression of unique solute transporters, and each segment has been mapped to a precise axial

location relative to the somites that comprise the embryonic trunk (*Figure 1A*, bottom panel), which facilitates the analysis of pattern formation within the renal progenitor field (*Wingert et al., 2007*).

Chemical genetics is a powerful approach to study developmental events in the context of the whole organism, and the application of chemical genetics in the zebrafish has led to a number of valuable discoveries about the mechanisms of organogenesis in diverse tissues, including derivatives of the mesoderm (*Lessman, 2011*; *Poureetezadi and Wingert, 2013*). Therefore, we hypothesized that a chemical genetic screen could provide new insights about the identity of nephrogenesis regulators. To this end, we performed a chemical genetic screen using the Screen-Well ICCB Known Bioactives Library (Enzo Life Sciences), a collection that includes 480 compounds with known biological activities. Zebrafish embryos were collected from timed matings of wild-type (WT) adults, and then arrayed in 96-well plates for control (dimethyl sulfoxide, DMSO) or experimental treatment between 4 and 24 hpf (*Figure 1B*). At the 24 hpf stage, embryos were fixed for multiplex WISH analysis, during which they were assessed for expression of a set of genetic markers that distinguished alternating nephron segments within the pronephros, namely *wt1b* to directly label the P, *slc20a1a* to label the PCT, and *slc12a1* to label the DE (*Figure 1B*). Because these riboprobes stain alternating nephron segments, they enabled precise scoring as to whether exposure to each chemical led to an expansion or restriction of these distinct cell types (*Figure 1C*, *Figure 1—source data 1*).

In total, 16.25% (78/480) of ICCB bioactives were associated with nephron phenotypes (*Figure 1C*, *Figure 1—source data 1*). The effect of each compound was annotated as to whether the experimental dosage was associated with WT development, an expansion in segment(s) (P+, PCT+, DE+) or a restriction in segment(s) (P-, PCT-, DE-) (*Figure 1C*, *Figure 1—source data 1*). The compounds that led to alterations in nephrogenesis included numerous RA pathway agonists and antagonists, such as 4-hydroxyphenylretinamide (4-HPR), a synthetic analog of all-trans RA (*Figure 1D*) (*Poureetezadi et al., 2014*). Compared to WTs, exposure to 1 mM 4-HPR led to an expansion of the PCT, caudal shift of the DE, and a dramatic expansion of the interval between these segments where the PST normally emerges, suggestive of an expanded PST segment (*Figure 1D*) (*Poureetezadi et al., 2014*). The observation that molecules which impact the RA pathway were flagged as hits in the screen provided an important positive control for our experimental system, given the well-established effects of RA levels on renal progenitors (*Wingert et al., 2007*; *Wingert and Davidson, 2011*; *Li et al., 2014*; *Cheng and Wingert, 2015*; *Marra and Wingert, 2016*; *Drummond et al., 2017*).

In further surveying the identities and respective classifications of the small molecules that impacted nephrogenesis, we noted a striking trend with regard to prostaglandin pathway agonists and tubule segmentation. Among the screen hits, a series of prostaglandin cytokine moieties were independently flagged as modifiers of tubule segment formation, including $PGD_2$, $PGA_1$, $PGJ_2$, and $PGB_2$ (*Figure 1D*, *Figure 1—source data 1*). Exposure to these bioactive prostaglandins was associated with changes in the pronephros whereby there was a reduced PCT segment length and a posterior shift in the position of the DE, such that there was a noticeably longer domain between these segment regions compared to WT control embryos (*Figure 1D*). The discovery that exposure to exogenous prostaglandins was linked with several segmentation changes was particularly fascinating to us because $PGE_2$ signaling has been associated recently with the development of several tissues, including HSCs and fate choice in endoderm derivatives between the liver and pancreas (*North et al., 2007*; *Nissim et al., 2014*). Therefore, we next sought to further explore how elevated prostaglandin levels, including $PGE_2$, affected nephron segment development.

## Elevated $PGE_2$ or $PGB_2$ levels induce an expansion of the PST segment and DL reduction

Prostaglandins typically have a short half-life, and have been characterized as secreted molecules that activate receptors close to their site of production, thus inducing local effects in a paracrine or autocrine fashion (*Smyth et al., 2009*). Prostaglandins have also been shown to elicit dosage-specific effects, leading to their description as morphogens (*Nissim et al., 2014*). Therefore, to validate and further explore the screening results, we selected two prostanoids: one was a hit from our screen, $PGB_2$, and the other was 16,16-dimethyl-$PGE_2$ (dm$PGE_2$), a long-acting derivative of $PGE_2$ which has been extensively used to study the effects of $PGE_2$ in zebrafish (*North et al., 2007*; *Goessling et al., 2009*; *Nissim et al., 2014*). WT embryos were collected and incubated in varying concentrations of drug (30 μM, 50 μM, 80 μM or 100 μM) from the 4 hpf stage to the 24 hpf stage. Double WISH was

then performed to determine the resultant nephron segments alongside the trunk somites, and the absolute lengths of nephron segment domains were also measured (*Figure 2*, *Figure 2—figure supplements 1* and *2*).

Exposure to dmPGE$_2$ or PGB$_2$ resulted in a dose-dependent increase in the domain length of the PST segment compared to WT embryos, visualized by WISH with the marker *trpm7* (*Figure 2A,B*, *Figure 2—figure supplements 1,2*). In conjunction with this change, the DL segment was significantly reduced in length, as visualized by WISH with the marker *slc12a3* (*Figure 2A,C*, *Figure 2—figure supplements 1,2*). Additionally, the rostral domain of the PCT was reduced in a dosage-dependent fashion, based on expression of *slc20a1a* (*Figure 2A,D*, *Figure 2—figure supplements 1,2*). Further, there was a caudal shift in the position of the DE segment though its absolute length was unchanged, based on expression of the DE-specific marker *slc12a1*, which resides between the domains of the PST and DL segments (*Figure 2A,D*, *Figure 2—figure supplements 1,2*). Overall, these results recapitulated the phenotypes observed following treatment with various bioactive prostaglandins during the chemical screen (*Figure 1D*). To determine if embryo dimensions were a factor in pronephros segment domain changes, we measured control and treated embryos from tip to tail as well as their pronephric domain (somite 3 to somite 18). We found no statistical differences in the body axis length or pronephric domain between WT controls and dmPGE$_2$ treated embryos (*Figure 2—figure supplement 3*). To further gauge the possible side effects of dmPGE$_2$ treatment on surrounding tissues, we assessed development of specific tissues using WISH. We noted no significant changes in the vascular marker *flk1* or primitive blood precursors using the marker *gata1* between WT controls and 100 µM dmPGE$_2$ treated embryos (*Figure 2—figure supplement 4A,B*). Furthermore, we performed o-dianisidine staining, which labels hemoglobinized erythrocytes and thereby provides a sensitive assessment of defects in circulation or vascular integrity that can be undetected by live imaging with stereomicroscopy. o-dianisidine staining showed that blood flow in WTs and 100 µM dmPGE$_2$ treated embryos was equivalent through the 48–55 hpf stage, as we did not observe compromised vessel integrity or hematomas (e.g. bleeding, blood pooling) (*Figure 2—figure supplement 4C*). This suggests that PGE$_2$ exposure did not cause major aberrations in tissues surrounding the pronephros. In sum, these observations confirmed the finding from the chemical screen that exogenous PGB$_2$ had profound effects on nephron segment formation, and revealed that alterations in PGE$_2$ had similar consequences.

## Expression of Ptges enzymes is required for normal distal pronephros segment development

Next, we determined whether endogenous prostaglandin biosynthesis mediated by the Ptges (e.g. Cox1, Cox2) enzymes was necessary for normal nephron segmentation. To test this, we incubated WT embryos with the compound indomethacin, a nonselective Cox1 and Cox2 enzyme inhibitor, which inhibits the first stage of prostanoid biosynthesis, and has been shown to suppress PGE$_2$ production in zebrafish by mass spectrometry (*Figure 1E*, *Figure 3*) (*Grosser et al., 2002*; *Cha et al., 2005*; *North et al., 2007*). Exposure of WT embryos to 30 µM indomethacin was associated with normal proximal segment locations along the embryonic trunk (*Figure 3A*, *Figure 3—figure supplement 1*). However, the balance of distal segments was disrupted after indomethacin treatment, such that the majority of embryos developed an *slc12a1*-expressing DE segment that was significantly expanded in length and an *slc12a3*-expressing DL segment that was significantly reduced in length compared to wild-type controls (*Figure 3A,B and C*). Absolute segment length measurements of the proximal domains in indomethacin treated embryos compared to wild-types confirmed there was no significant change in the lengths of these segments (*Figure 3D*, *Figure 3—figure supplement 1*). As with dmPGE2 treated embryos, we assessed the effect of indomethacin exposure at this dosage with various morphological dimensions and the formation of surrounding tissues such as the vasculature, and observed no differences compared to WT controls (*Figure 2—figure supplements 3,4*).

To further explore these results, we examined the effect of other small molecules that have been validated to interfere with Cox enzyme activity. Treatment with the Ptgs1 (Cox1) selective inhibitor SC-560 or the Ptgs2a (Cox2) selective inhibitor NS-398 (*Grosser et al., 2002*; *Cha et al., 2005*; *North et al., 2007*) induced an expansion of the DE segment and a restriction of the DL compared to wild-type embryos, while having no discernible effect on proximal segment development (*Figure 4*, *Figure 4—figure supplements 1* and *2*). The DE and DL segment domain phenotypes

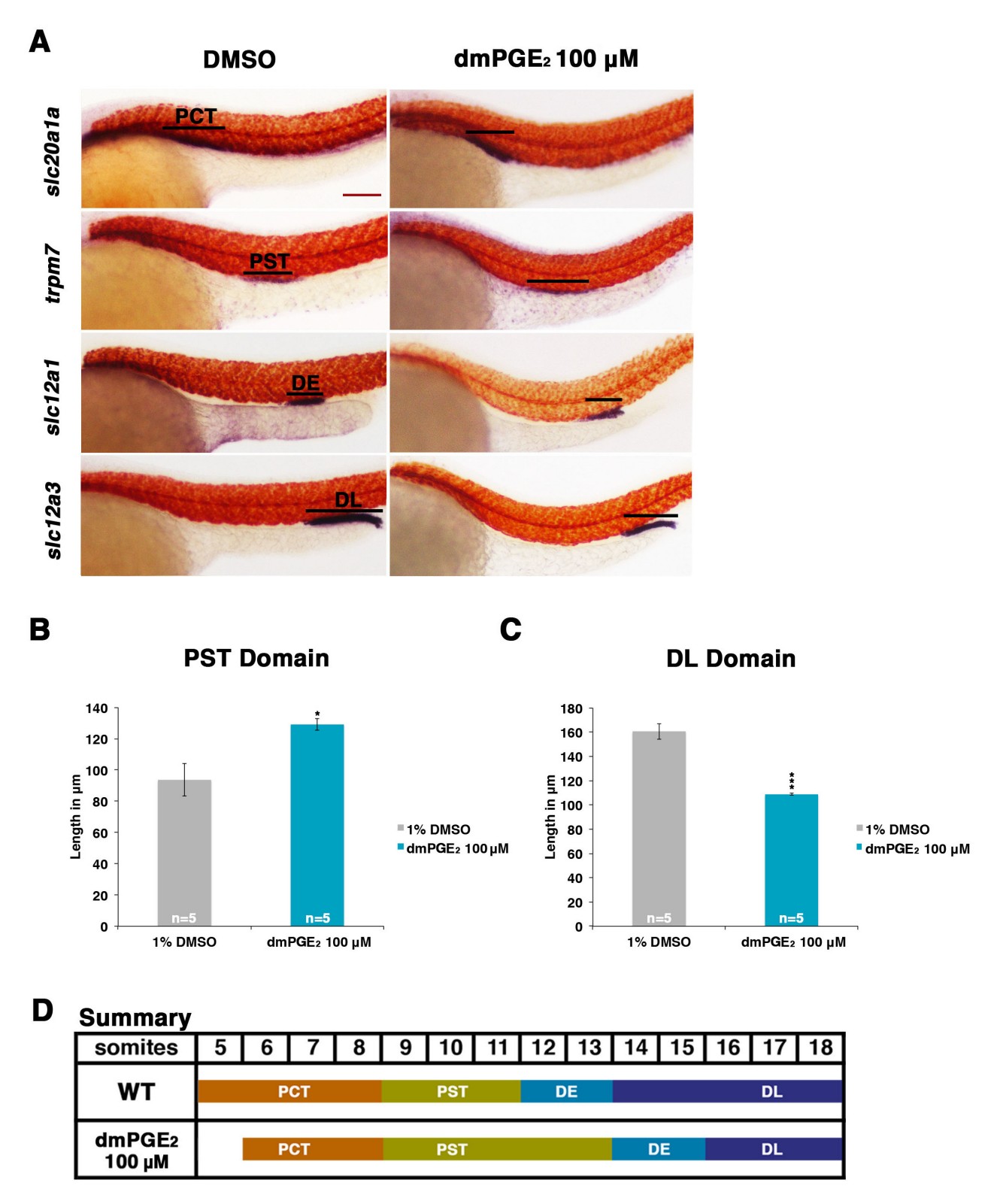

**Figure 2.** Exogenous prostaglandin activity promotes proximal straight tubule identity. (**A**) Embryos were exposed to 100 μM dmPGE$_2$ between 4 hpf and 24 hpf. WISH was used to stain for the PCT (*slc20a1a*), PST (*trpm7*), DE (*slc12a1*), and DL (*slc12a3*) (purple) and the somites (*smyhc1*) (red) at the 24 hpf stage. Black bars indicate segment gene expression domain. Red scale bar, 70 μm. (**B,C**) The PST and DL segments were measured in microns after
*Figure 2 continued on next page*

*Figure 2 continued*

incubation in 100 µM of dmPGE$_2$ (n = 5 for each control and experimental group). (**D**) Summary depicting the nephron segments after exogenous dmPGE$_2$ treatment. Data are represented as ± SD, significant by t test comparing each drug treatment to the DMSO control, *p<0.05, ***p<0.0005.

The following figure supplements are available for figure 2:

**Figure supplement 1.** Exogenous PGB$_2$ treatment is sufficient to expand the proximal straight tubule.

**Figure supplement 2.** dmPGE$_2$ and PGB$_2$ treatment have dosage-dependent effects on pronephros segmentation.

**Figure supplement 3.** Embryos dimensions remain largely unchanged by dmPGE$_2$ or indomethacin treatment.

**Figure supplement 4.** Perturbation of the prostaglandin pathway fails to elicit gross blood or blood vessel abnormalities during development.

following SC-560 or NS-398 treatment were statistically significant based on absolute length analysis compared to wild-type controls (*Figure 4—figure supplement 9*).

To corroborate the effects we observed on zebrafish distal nephron segment development from Cox1/2 enzyme inhibition, we generated *ptgs* knockdowns through microinjection of previously described translation blocking morpholinos to target *ptgs1*, *ptgs2a*, or both *ptgs1/2a* into 1-cell stage WT embryos (*Grosser et al., 2002*; *North et al., 2007*). Nephron segmentation was analyzed at the 24 hpf stage by WISH with the panel of specific markers to delineate the domains of the PCT, PST, DE and DL (*slc20a1a, trpm7, slc12a1* and *slc12a3*, respectively) as well as the somites (*smyhc1*). Single and double *ptgs1/2a* morphants developed a larger DE domain that was increased in length compared to WT control embryos (*Figure 4B–4D*, *Figure 4—figure supplements 1*, *2* and *9*). In addition, single and double *ptgs1/2a* morphants had a shortened DL segment compared to the DL in WT control embryos (*Figure 4B–4D*, *Figure 4—figure supplements 1*, *2* and *9*). These DE and DL segment domain phenotypes were all statistically significant based on absolute length analysis compared to WT controls (*Figure 4—figure supplement 9*). In contrast, PCT and PST segment development was normal in *ptgs1*, *ptgs2a*, or *ptgs1/2a* deficient embryos at the 24 hpf stage (*Figure 4—figure supplements 1* and *2*).

To further validate these findings, we conducted independent analysis of *ptgs1* or *ptgs2a* morpholinos that were confirmed to interfere with mRNA splicing, by which inclusion of an intron was found to generate transcripts encoding prematurely truncated proteins (*Figure 4—figure supplement 10*). Compared to wild-type controls, both of these *ptgs1* and *ptgs2a* splice morpholinos similarly affected pronephros development by causing a DE expansion and a DL reduction, with no perceivable consequence to the proximal segments (*Figure 4—figure supplement 3*). Again, the segment domain findings were statistically significant based on absolute segment length analysis with WT controls (*Figure 4—figure supplement 9*). Taken together, these data provide independent validation that expression of both *ptgs1* and *ptgs2a* are critical for normal formation of the DE and DL segments in the zebrafish pronephros.

## PGE$_2$ rescues nephron development in Ptgs-deficient zebrafish embryos

Next, we tested whether distal nephron segmentation in *ptgs1* and *ptgs2a* deficient embryos could be rescued by provision of a bioactive prostanoid. For these experiments, we selected dmPGE$_2$ treatment, in part to test the hypothesis that PGE$_2$ signaling is required for pronephros development. WT embryos were injected with *ptgs1* and *ptgs2a* morpholinos and then subsequently treated with 50 µM dmPGE$_2$ between the 4 hpf stage and the 24 hpf stage. Nephron segments were then assessed by WISH using our panel of segment-specific riboprobes. We observed that the alterations in DE and DL segments in *ptgs1* and *ptgs2a* deficient embryos were indeed rescued by exposure to dmPGE$_2$, a treatment combination that was again not associated with altered proximal segment domains (*Figure 4B,C*, *Figure 4—figure supplements 1*, *2* and *9*). Notably, the *ptgs1* and *ptgs2a* morphants treated with dmPGE$_2$ exhibited statistically similar DE and DL segment lengths compared to WT controls, in contrast to the longer DE and shortened DL in *ptgs1* and *ptgs2a* morphants treated with DMSO vehicle (*Figure 4—figure supplement 9*). These data indicate that the

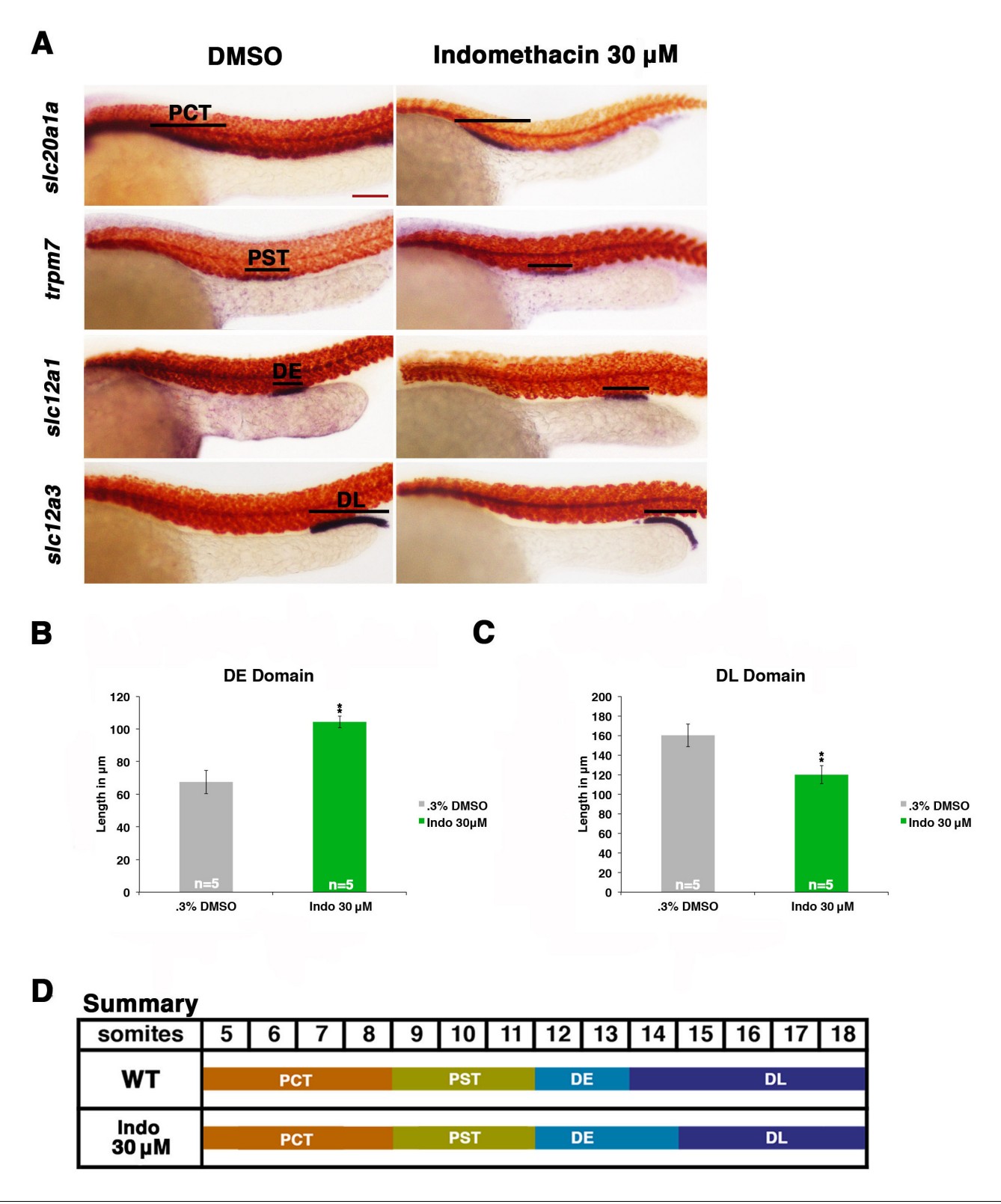

**Figure 3.** Prostaglandin production is required for normal distal cell fate specification. (**A**) Embryos were treated with 0.3% DMSO or the nonselective Cox inhibitor indomethacin at 30 μM from 4 hpf to 24 hpf. WISH was used to stain for the PCT (*slc20a1a*), PST (*trpm7*), DE (*slc12a1*), and DL (*slc12a3*) (purple) and the somites (*smyhc1*) (red) at the 24 hpf stage. Black bars indicate segment gene expression domain. Red scale bar, 70 μm. (**B,C**) The DE

*Figure 3 continued on next page*

*Figure 3 continued*

and DL segments were measured in microns (n = 5 for each control and experimental group). (**D**) A summary depicting the nephron segments after indomethacin treatment. Data are represented as ± SD, significant by t test comparing each drug treatment to the DMSO control group, **p<0.005.

The following figure supplement is available for figure 3:

**Figure supplement 1.** Indomethacin treatment has no effect on proximal cell-fate choice, yet shifts the balance between distal identities.

expansion of the DE segment and the restriction of the DL in *ptgs1* and *ptgs2a* deficient embryos were caused specifically by diminished prostaglandin activity, and implicate $PGE_2$ as the essential bioactive prostanoid because $dmPGE_2$ was sufficient to rescue pronephros segmentation in the context of either Cox1 or Cox2 knockdown.

## $PGE_2$ regulates nephron segmentation via the *ptger2a* and *ptger4a* receptors

$PGE_2$ is known to signal to its target cells by binding with the G-protein coupled receptor Prostaglandin E receptor 2a or the Prostaglandin E receptor 4a (Ptger2a, Ptger4a; also known as EP2 and EP4, respectively) (*Cha et al., 2006*). $PGE_2$ signaling in zebrafish acts through both Ptger2a and Ptger4a to modulate HSC formation (*North et al., 2007*), and during endoderm specification, wherein the differential expression of these receptors mediates tissue development at discrete stages (*Nissim et al., 2014*). There has been some spatiotemporal expression analysis of these genes during zebrafish embryogenesis as well, which revealed that *ptger2a* transcripts were expressed at the six somite stage (ss) within bipotential endoderm progenitors and that *ptger4a* transcripts were expressed at 72 hpf within liver precursors (*Nissim et al., 2014*). However, further characterization of these genes' expression in relation to other organs, such as the kidney, has not been addressed.

Based on this, we examined if Ptger2a and/or Ptger4a expression was associated with any stages of pronephros development. We utilized WISH to determine the spatiotemporal expression of *ptger2a* and *ptger4a* transcripts between the tailbud stage and 24 hpf to determine if they localized to the areas occupied by the nephron progenitors (*Figure 4—figure supplement 4*). Interestingly, we found that *ptger2a* transcripts were expressed in a continuous stretch of IM renal progenitors between the 12 ss and 24 ss based on their location in bilateral stripes of cells situated adjacent to the paraxial mesoderm (*Figure 4—figure supplement 4A*). *ptger4a* transcripts were similarly expressed in the IM renal fields between the 12 ss and 15 ss, where cells expressed varying levels of signal, suggesting patches of somewhat variable expression (*Figure 4—figure supplement 4B*). *ptger4a* transcripts showed low levels of ubiquitous mesoderm expression at the 20 ss, and then were localized to the cloaca region at the 24 ss (*Figure 4—figure supplement 4B*). We next performed double WISH in WT embryos at the 14 ss and 18 ss to label *cadherin17* (*cdh17*) expressing renal progenitors along with either *ptger2a* or *ptger4a* (*Figure 4—figure supplement 4C*). As expected, *ptger2a* expressing cells in the IM fully occupied the bilateral stripes of *cdh17* expressing cells, as did *ptger4a*, though again we noted the slight variability of *ptger4a* transcript staining in cells residing within the *cdh17* pronephros fields (*Figure 4—figure supplement 4C*). These data were consistent with the notion that Ptger2a and/or Ptger4a may operate in renal progenitors to modulate their development.

To test the hypothesis that Ptger2a and/or Ptger4a function was necessary for pronephros segmentation, we next utilized previously published 5'UTR or start site targeting morpholinos to abrogate expression of either *ptger2a* or *ptger4a* transcripts during embryogenesis (*Cha et al., 2006*; *North et al., 2007*), as well as independent pharmacological treatments with two different Ptger2a receptor antagonists to block its activity (*Figure 4E,F*, *Figure 4—figure supplements 2* and *5–9*). Deficiency of *ptger2a* or *ptger4a* resulted in a statistically significant expansion of the DE segment and a reduction of the DL (*Figure 4E,F*, *Figure 4—figure supplements 2* and *5–9*). These data recapitulate the phenotypic effects that resulted from treatment with the Ptgs1/2a small molecule inhibitors, as well as deficiency of Ptgs1, Ptgs2a, and the combination of Ptgs1/2a (*Figure 4E*). We also specifically evaluated whether *ptger2a* or *ptger4a* knockdown could be rescued by

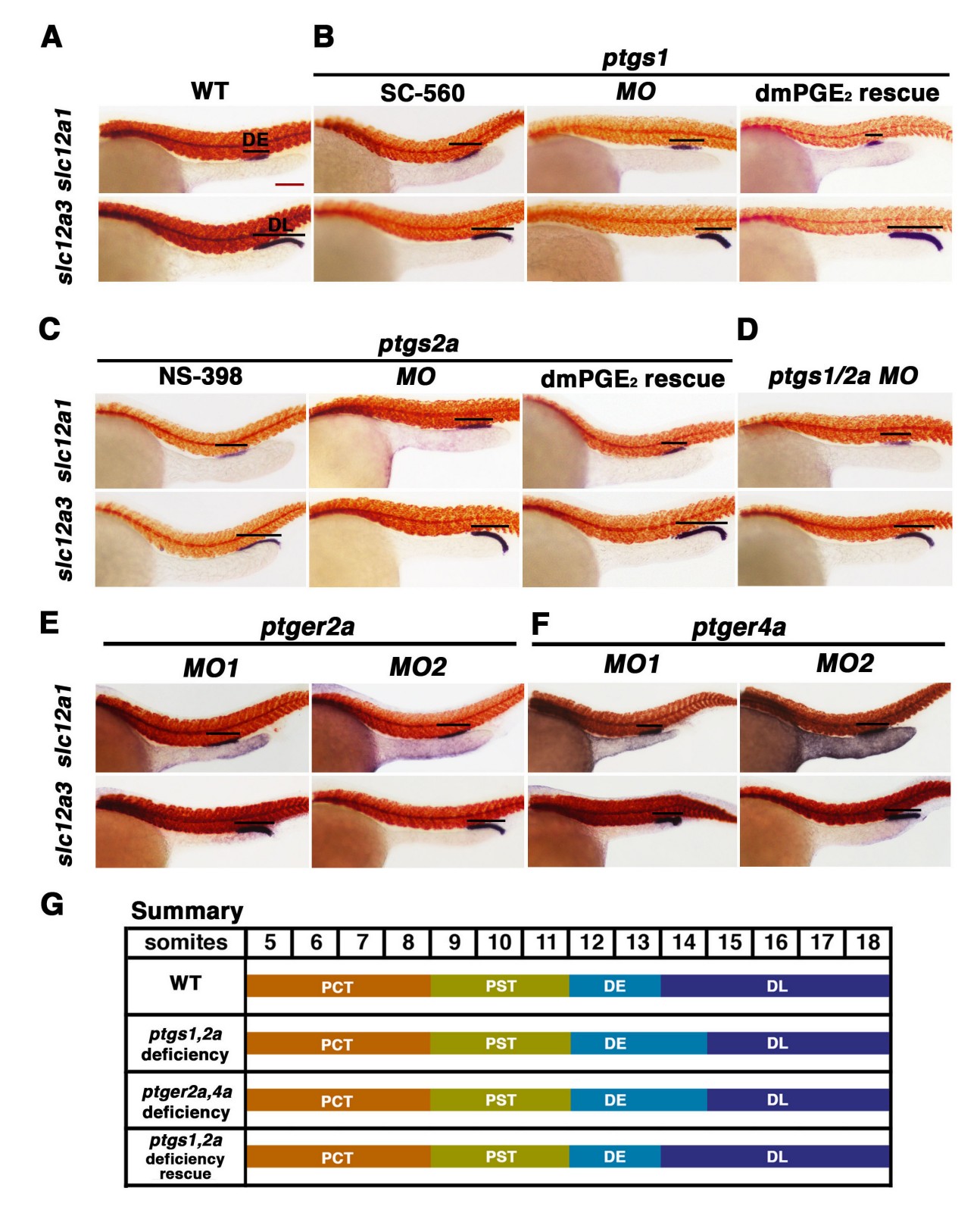

**Figure 4.** Prostaglandin signal inhibition results in an expansion of the distal early domain, which is rescued by the addition of exogenous dmPGE₂. Embryos were treated with a (**A**) 1% DMSO control, the (**B**) Ptgs1 selective inhibitor SC-560 at 50 µM from 4 hpf to 24 hpf, microinjected with the *ptgs1* MO alone, or microinjected with the *ptgs1* MO and treated with dmPGE₂ from 4 hpf to 24 hpf. Embryos were treated with the (**C**) Ptgs2a selective inhibitor NS-398 at 50 µM from 4 hpf to 24 hpf, microinjected with the *ptgs2a* MO or microinjected with the *ptgs2a* MO and then treated with dmPGE₂

*Figure 4 continued on next page*

*Figure 4 continued*

at 50 µM from 4 hpf to 24 hpf. (**D**) Embryos were microinjected with a combination of the *ptgs1* MO and *ptgs2a* MO. Embryos were microinjected with (**E**) *ptger2a* MO1, *ptger2a* MO2, (**F**) *ptger4a* MO1 and *ptger4a* MO2. (**A**–**E**) WISH was used to stain for the DE (*slc12a1*), DL (*slc12a3*) (purple), and the somites (*smyhc1*) (red) at the 24 hpf stage. Black bars indicate segment gene expression domain. Red scale bar, 70 µm. (**G**) Summary depicting the nephron segments after inducing deficiency of prostaglandin synthesis or receptor activity.

The following figure supplements are available for figure 4:

**Figure supplement 1.** Inhibition of Ptgs1 or Ptgs2a did not alter proximal segment identity.

**Figure supplement 2.** Diminishing Ptgs1 or Ptgs2a function causes distal early cell-fate identity to be favored at the expense of the distal late domain.

**Figure supplement 3.** Inhibiting prostaglandin synthesis with splice morpholinos promotes distal early identity.

**Figure supplement 4.** The prostaglandin receptor transcripts *ptger2a* and *ptger4a* are expressed in the pronephros during development and co-localize with the nephron.

**Figure supplement 5.** Morpholino inhibition of *ptger2a* fails to affect proximal segment identity.

**Figure supplement 6.** Inhibition of *ptger2a* using small molecule antagonists promotes distal early fate-choice.

**Figure supplement 7.** *ptger4a* MO knockdown results in an expansion of the distal early segment.

**Figure supplement 8.** Exogenous dmPGE$_2$ is incapable of rescuing distal cell fates in Ptger2a or Ptger4a deficient morphant embryos.

**Figure supplement 9.** Inhibiting PGE$_2$ production results in an expanded distal early domain at the expense of the distal late segment, which can be rescued by exogenous dmPGE$_2$.

**Figure supplement 10.** Validation of transcriptional changes using splice morpholinos to target *ptgs1*, *ptgs2a*, *ptger2a*, and *ptger4a*.

dmPGE$_2$, and found that dmPGE$_2$ was not sufficient to rescue either the DE segment expansion or DL reduction (*Figure 4—figure supplement 8*). These findings are consistent with the notion that PGE$_2$ acts specifically via Ptger2a and Ptger4a to mitigate DE-DL formation during pronephros ontogeny (*Figure 4—figure supplement 8*).

To further validate these observations and conclusions, we next examined pronephros segment development following morpholino-mediated knockdowns that were confirmed to alter the normal splicing of either *ptger2a* or *ptger4a* transcripts, and were consequently predicted to disrupt normal protein expression (*Figure 4—figure supplement 10*). We observed a statistically significant decrease in the length of the DL using these morpholinos, consistent with our previous observations with other knockdown reagents and pharmacological inhibitions, and thus lending further credence to the conclusion that DE and DL segmentation is reliant on Ptger2a or Ptger4a expression (*Figure 4—figure supplement 10*). Taken together, these data suggest that Ptger2a and Ptger4a have developmental roles in renal progenitors where they interact with PGE$_2$ to regulate distal nephron segment formation.

## Alterations in Ptgs activity or PGE$_2$ levels influences nephron segmentation after gastrulation

Since we found that *ptger2a* and *ptger4a* transcripts were expressed within renal progenitors beginning as early as the 12 ss, we hypothesized that prostaglandin signaling may begin to operate at that time period to influence pronephros segmentation. To test this, we treated WT embryos with either the nonselective Cox1/2 antagonist indomethacin (30 µM) to block Ptgs activity or the agonist dmPGE$_2$ (100 µM) from the 12 ss through to the 24 hpf time point, and then performed WISH to assess the pronephros segments. Indomethacin treatment during this time window elicited an expansion of the DE domain and a reduction of the DL similar to that seen from indomethacin treatments from 4 hpf to 24 hpf (*Figure 5A*). Further, dmPGE$_2$ treatment from the 12 ss to 24 hpf was

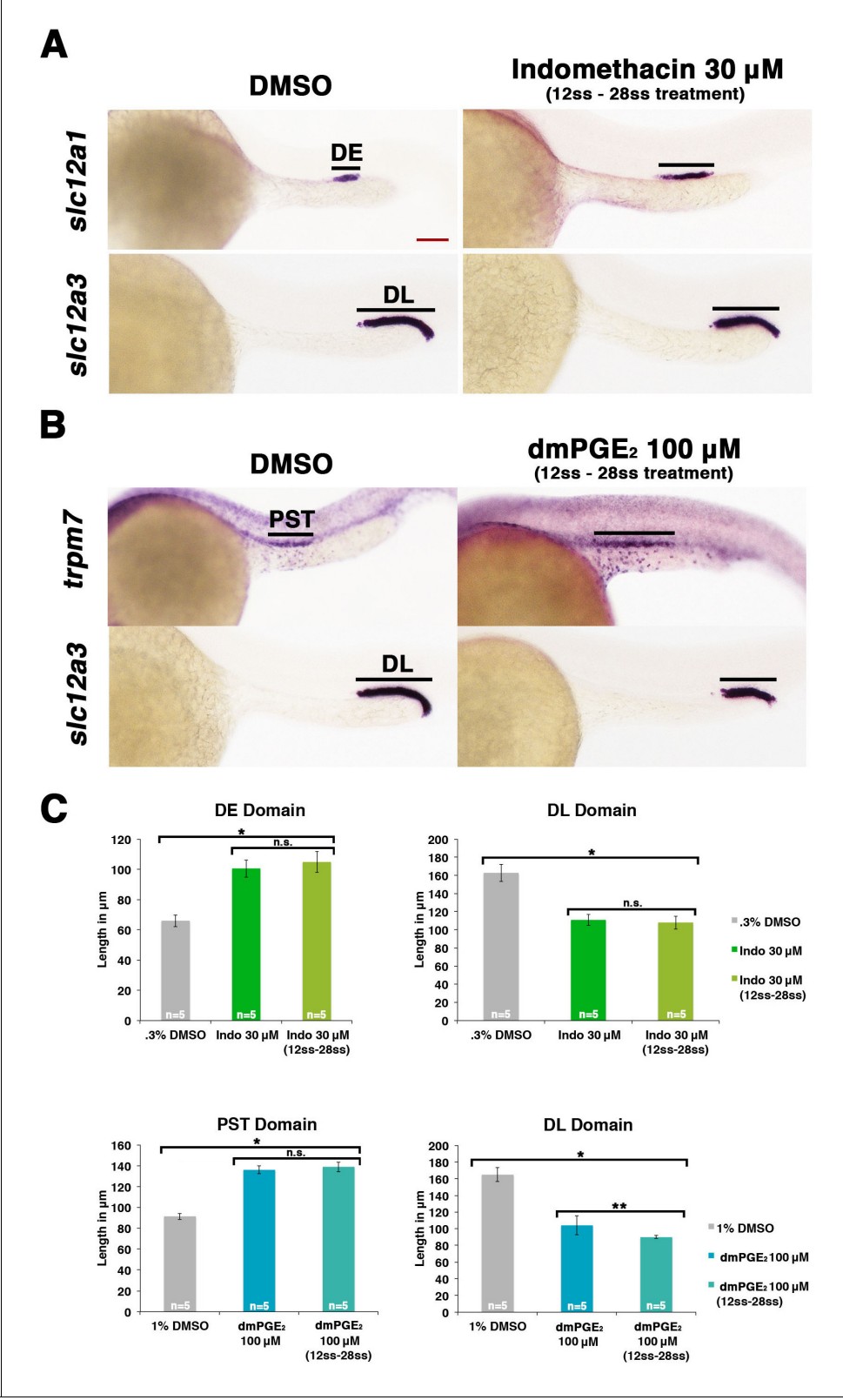

**Figure 5.** Prostaglandin signaling influences nephron patterning after gastrulation. Embryos were treated with 0.3% DMSO or 1% DMSO vehicle control, (**A**) 30 μM indomethacin, or (**B**) 100 μM dmPGE$_2$ from the 12 ss to the 28 ss and stained for the PST (*trpm7*), DE (*slc12a1*), and DL (*slc12a3*) (purple) using WISH at the 28 ss stage. Black bars indicate segment gene expression domain. Red scale bar, 70 μm. (**C**) The PST, DE, and DL domains
*Figure 5 continued on next page*

*Figure 5 continued*

were measured in microns (n = 5 for each control and experimental group). Data are represented as ± SD, ANOVA used to compare samples, *p<0.01, **p<0.05, where n.s. indicates not significant.

The following figure supplement is available for figure 5:

**Figure supplement 1.** Inhibiting Ptger2a after gastrulation induces an expansion of the distal early and a restriction of the distal late segment.

sufficient to induce an expansion of the PST and a reduction of the DL (*Figure 5B*). Absolute measurements of these segments changes in indomethacin and dmPGE$_2$ treated embryos revealed that they were significant compared to controls and were similar to pharmacological exposures performed between the 4 hpf and 24 hpf time period (*Figure 5C*). Interestingly, we also found that treatments with either of two different Ptger2 small molecule antagonists, PF04418948 or AH6809, from the 12 ss to 24 hpf was likewise sufficient to induce a statistically significant expansion of the DE segment and reduction of the DL segment compared to WT controls (*Figure 5—figure supplement 1*). These data identify the 12 ss through the 24 hpf time period as the critical interval when pronephros progenitors require PGE$_2$ signaling for normal segment development.

## Cox expression and PGE$_2$ levels affect renal progenitor transcription factor domains

Several transcription factors are known to be critical for proper tubule segment patterning during zebrafish pronephros development. Some of these include: *sim1a*, which is expressed throughout both the PCT and PST, and is essential for PST fate (*Cheng and Wingert, 2015*); *irx3b*, which is expressed throughout both the PST and DE, and is essential for DE segment fate (*Wingert and Davidson, 2011*; *Morales and Wingert, 2014*); and *mecom*, which is expressed dynamically along the renal progenitor field, ultimately becoming restricted to the DL domain where it is essential for normal formation of this segment (*Li et al., 2014*). Given these genes' respective roles in segment ontogeny, we hypothesized that PGE$_2$ signaling affects renal progenitor fate by modulating the expression domains of one or more of these crucial factors. To investigate this, WT embryos were treated with either a control vehicle DMSO, dmPGE$_2$, or indomethacin between the 4 hpf stage and the 20 ss, and then the spatial distribution of *sim1a*, *irx3b*, or *mecom* transcripts in the IM renal progenitors was assessed by WISH (*Figure 6*, *Figure 6—figure supplement 1*).

Interestingly, we found that dmPGE$_2$ treatment led to a significant expansion of the *sim1a* and *irx3b* domains, along with a significant reduction of the *mecom* domain, in the majority of embryos (*Figure 6A,B*). These alterations are consistent with the observation that dmPGE$_2$ expands the PST segment and restricts the DL (*Figure 2*). Further, we found that indomethacin treatment led to no significant change in the *sim1a* domain, while the *irx3b* domain was significantly increased in absolute length and the *mecom* domain was significantly reduced in length (*Figure 6A,B*). These alterations are in keeping with the prior observations that nonselective Cox inhibition with indomethacin, selective Cox1 or Cox2 inhibitors, as well as knockdown or inhibition of Ptger2a and Ptger4a expanded the DE segment and reduced the DL (*Figures 3,4*). In summary, these data suggest that PGE$_2$ signaling influences segment programs in part by affecting the expression of *sim1a*, *irx3b*, and *mecom*, either directly or indirectly, to mediate nephron segmentation.

## Cox activity and PGE$_2$ act upstream of *irx3b* and *sim1a*

Next, we tested the epistatic relationships between prostaglandin signaling and the essential transcription factors *irx3b*, *mecom*, and *sim1a*. As knockdown of *irx3b* results in loss of the DE segment (*Wingert and Davidson, 2011*), we exposed *irx3b* morphants to the Ptgs1 (Cox1) selective inhibitor SC-560 to test how the combined deficiency of *irx3b* and prostaglandin synthesis would impact the process of DE development during nephron segmentation. Embryos that were treated with SC-560 concomitant with *irx3b* deficiency failed to form a DE segment, similar to *irx3b* deficiency alone (*Figure 7A,B*). Taken together with the observation that indomethacin treatment was sufficient to expand the *irx3b* expression domain, this result is consistent with the conclusion that prostaglandin signaling occurs upstream of *irx3b* to regulate DE segment development.

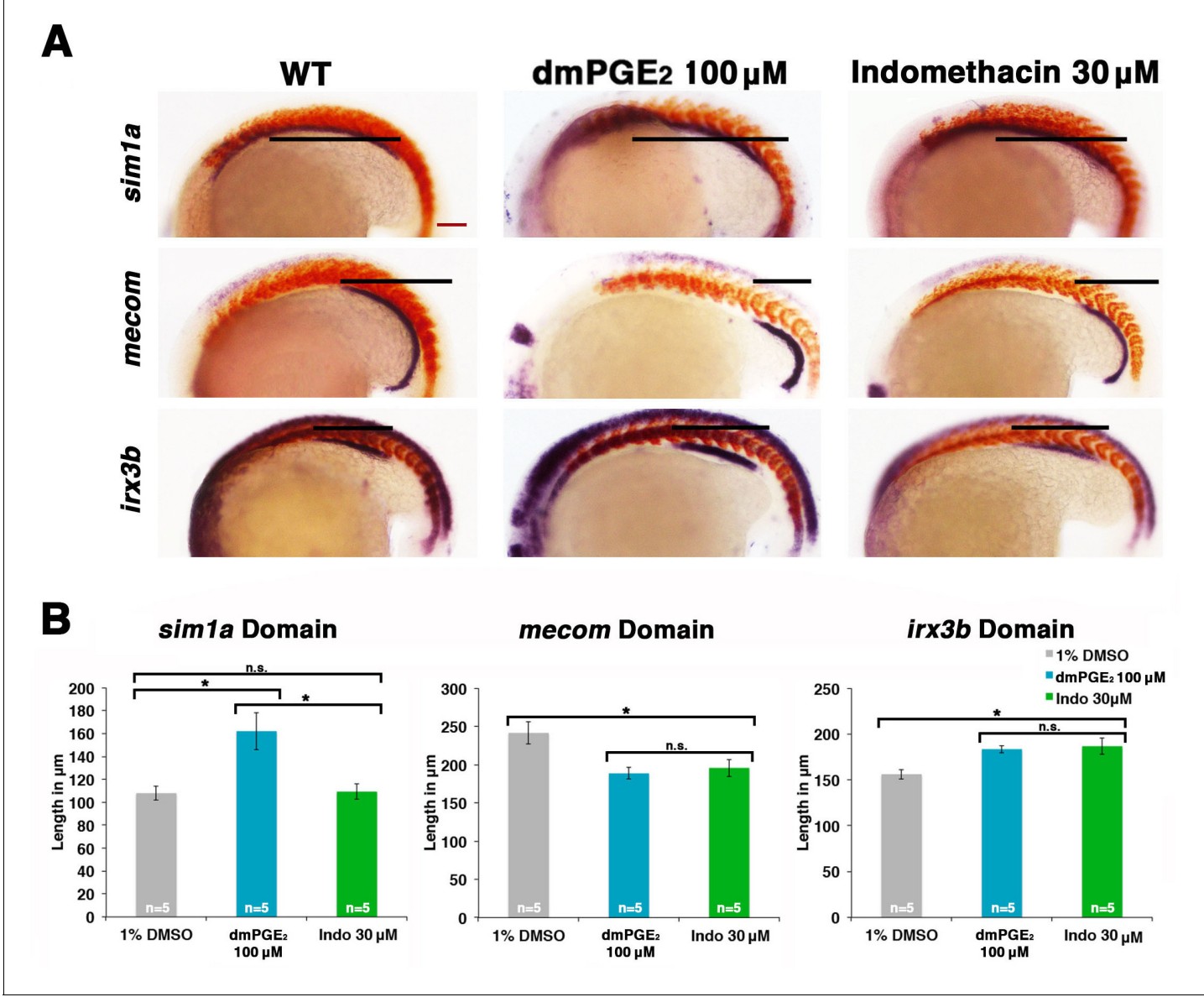

**Figure 6.** Prostaglandin signaling modulates key nephrogenesis transcription factors. (**A**) Embryos were treated with a 1% DMSO control, dmPGE₂ at 100 μM, or indomethacin at 30 μM from 4 hpf to the 20 ss. WISH was used to stain for the transcription factors *sim1a*, *mecom*, and *irx3b* (purple) and the somites (red) at the 20 ss stage. Black bars indicate segment gene expression domain. Red scale bar, 70 μm. (**B**) The *sim1a*, *mecom* and *irx3b* domains were measured in microns (n = 5 for each control and experimental group). Data are represented as ± SD, ANOVA used to compare samples, *p<0.01, where n.s. indicates not significant.

The following figure supplement is available for figure 6:

**Figure supplement 1.** Prostaglandin signaling alters the expression of transcription factors necessary for normal patterning of the nephron.

As we observed that the domain of *mecom* expression in renal progenitors is restricted when prostaglandin production was blocked with indomethacin (*Figure 6*), and that *mecom* deficiency in turn is associated with a reduced DL, we next explored whether restoration of *mecom* would be sufficient to rescue DL segment development in the absence of normal prostaglandin synthesis. To test this, *mecom* capped mRNA was synthesized and microinjected into 1-cell stage embryos, which were then treated with indomethacin or DMSO control as previously described (*Figure 7—figure supplement 1*). Overexpression of *mecom* at dosages ranging from 14–70 pg, in either the

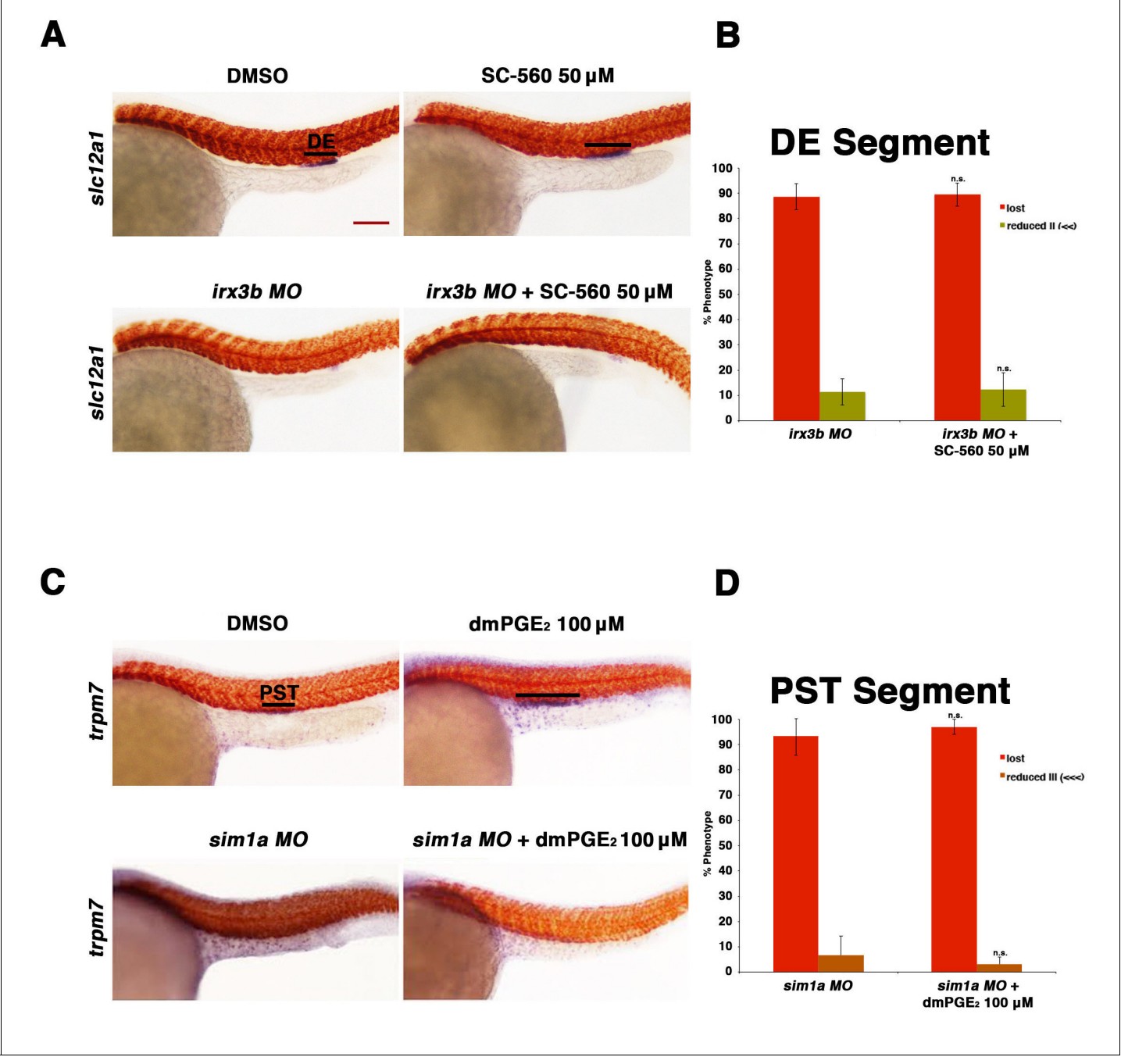

**Figure 7.** Prostaglandin signaling acts upstream of the transcription factors *sim1a* and *irx3b*. (**A**) Embryos were microinjected with the *irx3b MO* and treated with 0.5% DMSO or 50 μM SC-560 from 4 hpf to 24 hpf and stained for the DE using WISH. WT control embryos were also treated with 0.5% DMSO and 50 μM SC-560. (**B**) The DE (*slc12a1*) (purple) was stained for using WISH and quantified in triplicate, with at least 20 embryos per control and experimental group, by observing for the presence of the DE segment at the 24 hpf stage. Black bars indicate segment gene expression domain. Red scale bar, 70 μm. (**C**) Embryos were microinjected with the *sim1a MO* then treated with either 1% DMSO or 100 μM dmPGE$_2$ from 4 hpf to 24 hpf. 1% DMSO and 100 μM dmPGE$_2$ was also applied to WT control embryos from 4 hpf to 24 hpf. The PST (purple) and somites (red) were then stained for using WISH. (**D**) Quantification was undertaken in triplicate based on the presence of a PST or lack thereof, with at least 20 embryos per control and experimental group. Data are represented as ± SD, with t tests comparing each drug treatment to the corresponding DMSO control group, where n.s. indicates not significant.

The following figure supplements are available for figure 7:

**Figure 7—figure supplement 1.** Overexpression of *mecom* cRNA failed to rescue the distal late segment in indomethacin treated embryos.

*Figure 7 continued on next page*

*Figure 7 continued*

**Figure supplement 2.** Prostaglandin treatment shifts the distal early and restricts the distal late in *sim1a* morphants.

experimental or vehicle control group, was not sufficient to alter DL segment fate (*Figure 7—figure supplement 1*, data not shown). At higher dosages of *mecom* cRNA, embryos were dysmorphic, which has been reported previously (*Li et al., 2014*), precluding further examination with this approach. Despite these negative results, the alterations in the *mecom* expression domain in indomethacin treated embryos suggest that prostaglandin synthesis likely acts upstream of *mecom* to influence DL segment development.

Finally, we explored the relationship between *sim1a* and $PGE_2$ signaling. Previous work from our laboratory has demonstrated that *sim1a* overexpression is sufficient to expand the PST segment (*Cheng and Wingert, 2015*). In light of this, along with our present finding that exogenous $PGE_2$ treatment induces both an expanded PST segment and expanded *sim1a* domain, we hypothesized that the gain of function prostaglandin phenotype was reliant on *sim1a* for the alteration in PST fate. To test this, embryos were microinjected at the 1-cell stage with a *sim1a* (*Cheng and Wingert, 2015*), then incubated in either DMSO control or treated with $dmPGE_2$ between 4 and 24 hpf. We found that knockdown of *sim1a* concomitant with dmPGE2 treatment led to an abrogation of the PST segment, similar to *sim1a* deficiency alone (*Figure 7C,D*, *Figure 7—figure supplement 2*). These results are consistent with the conclusion that *sim1a* acts downstream of $PGE_2$ signaling in the context of exogenous treatment to drive expansion of the PST segment.

## Discussion

Understanding the genetic factors necessary to generate different cell types is an important aspect of developmental biology. Knowledge of these, along with an appreciation of the modulators that can impact the genesis of cell lineages, including their related morphogenetic processes, provides powerful insights relevant to congenital defects, disease pathology, regeneration and *in vitro* reprogramming (*Morales and Wingert, 2014*). Relevant to the present report, the kidney organ has many associated congenital diseases, and there is an escalating incidence of acute and chronic renal diseases for which a deeper understanding of mesodermal developmental processes has many possible applications (*Nakanishi and Yoshikawa, 2003*).

In this study, we uncovered evidence of a role for prostaglandin signaling in nephron segment formation during embryonic zebrafish development. Prostaglandins have diverse and potent biological actions (*Funk, 2001*), however, their effects on developing tissues including stem cells have only recently begun to be appreciated (*North et al., 2007*; *Goessling et al., 2009*; *Nissim et al., 2014*). By conducting a chemical genetic screen in zebrafish embryos to identify factors that affect nephron development, we found that several prostaglandin moieties were capable of modulating proximal-distal segmentation, first validating $PGB_2$ and then subsequently identifying $PGE_2$ as well (*Figure 8*). Specifically, we demonstrated that the addition of exogenous $PGE_2$ or $PGB_2$ was sufficient to increase PST segment size in a dosage-dependent manner. Using several chemical and genetic approaches, we demonstrated that abrogated prostaglandin activity alters formation of the DE and DL nephron tubule segments, where deficiencies in *ptgs1*, *ptgs2a*, or the $PGE_2$ receptors encoded by *ptger2a* and *ptger4a* led to an expanded DE and reduced DL segment, and that $PGE_2$ could specifically rescue the loss of Ptgs1 or Ptgs2a. We also determined that changes in Cox-mediated prostaglandin synthesis or $PGE_2$ correlated with alterations in the expression domains of essential segmentation transcription factors in renal progenitors, suggesting some mechanisms by which prostaglandin signaling acts to influence segmentation (*Figure 8*). Understanding how these changes relate with morphogenesis involving cellular dynamics (e.g. proliferation, turnover) or even migration of the renal progenitors will be important aspects for future investigations.

Previous studies have shown that $PGE_2$ is among the major prostanoids produced during the first day of zebrafish embryogenesis (*Cha et al., 2005*, *2006*). Based on this and our ability to rescue Cox1/2 deficiency with $dmPGE_2$ treatment alone, we theorize that $PGE_2$ is the central endogenous signaling component that affects pronephros development. Further, the localization of *ptger2a* and

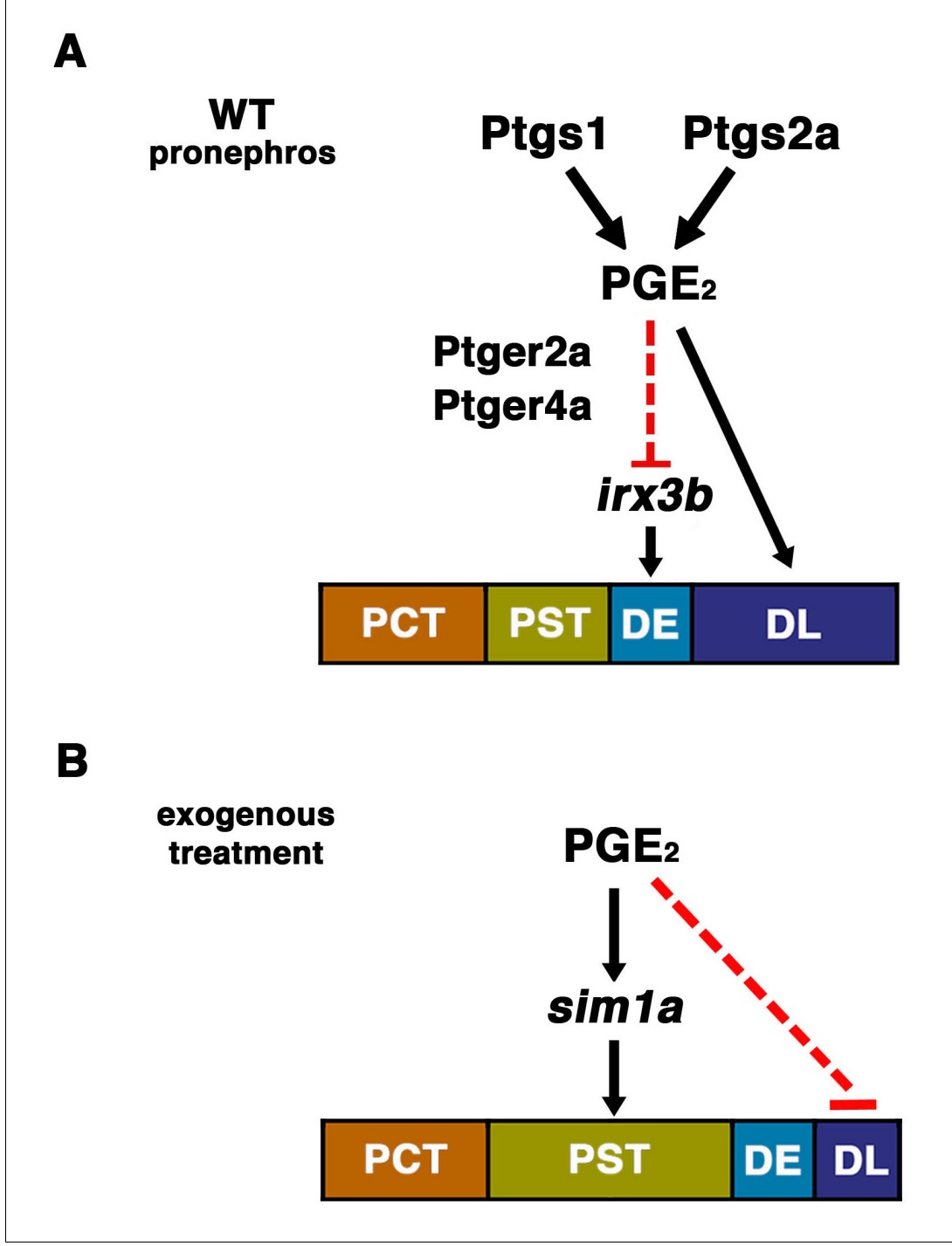

**Figure 8.** The role of prostaglandin signaling during zebrafish nephrogenesis. (**A**) PGE$_2$ is generated by the Ptgs1 and Ptgs2a enzymes, which interact with the Ptger2a and Ptger4a receptors to define the boundaries of the DE by negatively regulating the domain of *irx3b* expression and promoting DL identity. (**B**) Exogenous prostaglandin treatment promotes PST identity, partly by regulating *sim1a* expression, and these alterations occur at the expense of the DL segment. Not depicted: exogenous PGE$_2$ is also associated with an increased *irx3b* domain, which correlates with the shifted posterior location of the DE in the context of PST expansion. Further, in both (**A**) and (**B**), the *mecom* expression domain is reduced in length, which is likely part of the mechanism underlying the DL segment size reduction in these cases.

*ptger4a* expression in renal progenitors suggests that these cells directly receive and respond to prostaglandin signals. Since prostaglandins are known to act in an autocrine or paracrine fashion with short half-lives, we speculate that the IM renal precursors themselves or nearby tissues like the paraxial mesoderm are the local source(s) of prostanoids. Based on the results of several epistasis experiments, we currently hypothesize that $PGE_2$ signaling restricts the DE fate in part by negatively regulating *irx3b*, and that elevated $PGE_2$ levels expand the PST segment through positive regulation of *sim1a*, though future studies are needed to delineate if these interactions are direct or indirect (*Figure 8*). Finally, our data repository of small molecules that affect pronephros ontogeny will provide a useful starting point for future studies by our lab and others to study nephrogenesis further in zebrafish or in other vertebrates (*Figure 1—source data 1*).

## Mechanisms of prostaglandin signaling during nephrogenesis

Early genetic studies that interrogated the effect of disrupting prostaglandin synthesis in homozygous mutant COX-1 or COX-2 mice did not report observing overt abnormalities at birth (*Langenbach et al., 1995*; *Morham et al., 1995*; *Mahler et al., 1996*). Similarly, knockouts of other biosynthesis enzymes and the prostaglandin receptors had normal neonatal phenotypes (*Sugimoto et al., 2000*; *Kobayashi and Narumiya, 2002a*, *2002b*). Intriguingly, however, mice lacking COX-2 exhibit postnatal kidney pathologies associated with neonatal fatality, including nephron hypoplasia and atrophy, impaired cortical growth, and even cyst formation in multiple nephron segments—phenotypes suggestive of significant disruptions in renal ontogeny (*Dinchuk et al., 1995*; *Morham et al., 1995*; *Mahler et al., 1996*). These observations were not further explored until recently, however, where COX-2 gene dosage and pharmacological inhibition were linked to renal defects in glomerular size, such that COX-2$^{+/-}$ mice were found to exhibit kidney insufficiency (*Slattery et al., 2016*). In alignment with these data, exposure to COX inhibitors like indomethacin during human development is associated with renal failure, where the histological aspects include small glomeruli and microcystic lesions among other defects (*Gloor et al., 1993*; *van der Heijden et al., 1994*; *Kaplan et al., 1994*).

In light of these recent observations along with our findings, we propose that prostaglandin signaling, likely through $PGE_2$, has critical roles in nephrogenesis, which warrant further investigation. However, there are significant challenges of studying nephron formation in mammalian models due to their *in utero* development, and while metanephric organ culture has been informative for branding morphogenesis studies, it is not conducive to studying nephrogenesis. Therefore, the zebrafish pronephric model provides an alternative for continued genetic dissection of the cellular and molecular effects in nephron development due to alterations in prostaglandin levels. Based on the requirement for $PGE_2$ during pronephros development, it will be interesting to explore its effects on renal progenitors in other stages of zebrafish kidney ontogeny, and during new nephron formation and epithelial regeneration events in adults as well (*McCampbell et al., 2015*). In the near future, emergent organoid technologies will likely provide a complementary *in vitro* experimental system to probe the mechanisms of prostaglandin signaling in mammalian nephrogenesis (*Chambers et al., 2016*).

Further, while our report documents an essential role for $PGE_2$ signaling during nephron formation, more work needs to be done to further understand the genetic networks that affect segment fate. For example, we have previously shown that RA acts as a morphogen in the zebrafish, and that a gradient of RA induces proximalization of the impending pronephros (*Wingert et al., 2007*; *Wingert and Davidson, 2011*). It is intriguing to speculate that a prostaglandin signaling gradient, where $PGE_2$ acts as a morphogen, as recently proposed (*Nissim et al., 2014*), may articulate with RA to balance proximo-distal specification of the renal progenitors, though further studies are needed to interrogate this possibility. In addition, prostaglandins have been shown to initiate transcription through either interacting with cognate Ptger (EP) receptors or alternatively, passing through the cell membrane and binding with peroxisome proliferator-activated receptors (PPARs) (*Guan and Breyer, 2001*; *Berger and Moller, 2002*). Interestingly, PPARs can heterodimerize with Retinoid X Receptors (RXRs), a nuclear receptor for RA (*Guan and Breyer, 2001*; *Berger and Moller, 2002*). Also, it has been shown that different prostaglandins can interact interchangeably, at the right concentration, with the various Ptger receptors (*Tootle, 2013*). This might explain why different prostaglandins could induce an expansion of the PST.

## Understanding the broader roles of PGE$_2$ signaling in development

Prostaglandins have only just started to become recognized as important factors and determinants of cell fate decisions and growth during development. These new roles for prostaglandins have been revealed in part through a study showing that PGE$_2$ activity has a conserved function to expand the domain of HSCs in development and enhance their ability to home to the bone marrow in transplants (*North et al., 2007*). PGE$_2$ is currently in phase two clinical trials for increasing the efficiency of bone marrow transplants (*Hagedorn et al., 2014*). Furthermore, PGE$_2$ was shown to be a regulator of bipotential endoderm cell fate decisions in development for zebrafish and mouse endodermal cells (*Nissim et al., 2014*). Curiously, it was also shown that PGE$_2$ activity later in organ formation induced proliferation of both the liver and pancreas buds. This change in the function of PGE$_2$ is explained by the spatial and temporal expression of *ptger2a* and *ptger4a,* where blocking *ptger2a* promoted liver versus pancreas specification and blocking *ptger4a* promoted the outgrowth of the liver and pancreas buds. These cornerstone studies, along with the present report, give credence to the notion that PGE$_2$ is a key regulator of progenitor populations during embryogenesis and set the stage for further inquiry into how prostaglandin signaling affects developing cell populations. As more knowledge comes to light about how PGE$_2$ and other prostaglandins influence ontogeny, they are likely to become an increasingly intriguing option for clinical therapeutic applications.

# Materials and methods

## Zebrafish husbandry

Zebrafish were cared for and maintained in the Center for Zebrafish Research at the University of Notre Dame using experiments approved under protocol 16–025. Adult Tübingen strain fish were used for these studies, and their offspring were staged as described (*Kimmel et al., 1995*).

## Chemical genetic screen and other chemical treatments

Zebrafish wild-type (WT) embryos were arrayed and treated with small molecules using the ICCB Known Bioactives Library as described (*Poureetezadi et al., 2014*). Zebrafish embryos were staged at 2 hpf, then at least 30 fertilized embryos were arrayed into the chambers of 24-well plates and incubated at 28℃ in E3 media until just prior to 4 hpf. Working stocks of small molecules were stored at −80℃, then dissolved in 100% DMSO to make 10 mM concentrations (*Lengerke et al., 2011*). For drug exposure, the E3 media was completely drawn off the embryos using a glass transfer pipet and the appropriate solution of DMSO, PGB$_2$, dmPGE$_2$, indomethacin, SC-560, NS-398, AH6809, or PF04418948 was applied at a discrete development time point (eg 4 hpf or 12 ss) (American Bioanalytical, Enzo Life Sciences, Santa Cruz) (*North et al., 2007*; *Eisinger et al., 2007*; *Jin et al., 2014*). Embryos were raised to the 20 ss or 24 hpf, washed three times with E3, then fixed in 4% paraformaldehyde. For rescue of prostaglandin activity, *ptgs1 or ptgs2a* deficient embryos were generated, and cohorts of approximately 30 embryos were arrayed in 24-well plates with E3, then placed in a 28℃ incubator until 4 hpf. E3 was then completely drawn off the wells using a glass transfer pipet and a 50 μM concentration of dmPGE$_2$ was applied. The embryos were placed into a 28℃ incubator, raised until 24 hpf, washed three times with E3, and fixed as previously described.

## Embryo staining for WISH, o-dianisidine, and image acquisition

WISH was conducted as described (*Cheng et al., 2014*). RNA probes were digoxigenin or fluorescein labeled and generated by *in vitro* transcription using plasmid templates as described (*Wingert et al., 2007*; *Wingert and Davidson, 2011*; *Lengerke et al., 2011*; *Li et al., 2014*; *Cheng and Wingert, 2015*). For o-dianisidine staining, embryos were treated with 1% DMSO, 50 μM dmPGE$_2$, or 30 μM, were allowed to develop until 48 hpf and o-dianisidine staining was performed (*Wingert et al., 2004*). Images were taken using a Nikon Eclipse Ni with a DS-Fi2 camera. Figures were assembled using Adobe Photoshop CS5.

## Morpholino knockdown and RT-PCR

Antisense morpholino oligonucleotides (MOs) were purchased from Gene Tools, LLC. MOs were solubilized in DNase/RNase free water to create 4 mM stocks and stored at −20℃. WT embryos were collected after fertilization, injected with approximately 1 nl of diluted MO at the 1-cell stage and

then placed in a 28℃ incubator until the desired stage. MO sequences and dosages used were: *irx3b* 5'-ATAGCCTAGCTGCGGGAGAGACATG-3', 1 ng (*Wingert and Davidson, 2011*); *ptger2a* MO1 5'-GATGTTGGCATGTTTGAGAGCATGC-3', 3 ng (*North et al., 2007*); *ptger2a* MO2 5'-ACTG TCAATACAGGTCCCATTTTC-3', 1.6 ng (*North et al., 2007*); *ptger2a* MO3 splice 5'-CAATAAATC TTACTATTAACGGCAG-3', 3 ng; *ptger2a* MO4 splice 5'-ATGTACACACGGATCTG-AAGAGAAG-3', 3 ng; *ptger4a* MO1 5'-CGCGCTGGAGGTCTGGAG-ATCGCGC-3', 3 ng (*North et al., 2007*); *ptger4a* MO2 5'-CACGGTGGGCTCCATGCTGCTGCTG-3', 3 ng (*Cha et al., 2006*); *ptger4a* MO3 splice 5'-CCTGGAACTTACAACAAGCGGGATT-3', 3 ng; *ptger4a* MO4 splice 5'-TGAGAAACA-CC TGGACCTGCCAGAA-3', 3 ng; *ptgs1* MO 5'-TCAGCAAAAAGTTACACTCTCTCAT-3', 3 ng (*North et al., 2007*); *ptgs2a* MO 5'-AACCAGTTTATTCATTCCAGAAGTG-3', 3 ng (*Grosser et al., 2002*); *ptgs1* MO splice 5'-AACTTTCATTGCTC-ACCTCTCATTG-3', 2 ng; *ptgs2a* MO splice 5'-ATT-CAACTTA-CACAACAGGATATAG-3', 2 ng (*Yeh et al., 2009*), *sim1a* MO 5'-TCGACTTCTCCTTCA TGCTCTACGG-3', 1 ng (*Cheng and Wingert, 2015*). To assess knockdown efficacy, RT-PCR and sequence analysis was performed as previously described (*Marra and Wingert, 2016*) and using the following primers, where uppercase letters indicate location in an exon and lowercase indicates primer location in an intron: ptgs1-F1 5'-TTTATTTATTTGCAGCTTTTTCTT-3'; ptgs1-R1 5'-CAGTG TTTGATGAAGTCGGGCTTTC-3'; ptgs2a-F1 5'-CTGAGCTTCTCACACGCATCAAAT-3'; ptgs2a-R1 5'-GGCGAAGAAAGCAAACATGAGACT-3'; ptger2a-F1 5'-AGACCGAGCGTATGCCAATGT- 3'; ptger2a-R4 5'-caggagggctaataattcagactt-3'; ptger2a-F3 5'-ctgtttcagtgatcagtttgt-3'; ptger2a-R7 5'-CCGCAGAGCTATGAGATCAGTC-3'; ptger2a-R8 5'-GCTGAGGATGATGAACACCAAG-3'; ptger4a-F3 5'-ATGGTCATCCTGTTGATCGCC-3'; ptger4a-R2 5'-aatgagagtcctggaacttac-3'; ptger4a-F5 5'-gggtgtagtcatttatgttgagca-3'; ptger4a-R5 5'-CAGGACCGCTTTACGCAGTAAG-3'.

## Quantification of phenotypes, segment measurements, imaging and statistical analysis

Gene domains were assessed with respect to somite boundaries to assess pattern formation (*Wingert et al., 2007*). Segment domains were analyzed and counted in triplicate with at least 15 embryos per replicate. To measure absolute segment lengths, images were taken of at least five representative embryos. Images were collected using a Nikon Eclipse Ni with a DS-Fi2 camera and measurements performed with Nikon Elements Software. The average was generated and standard deviation (± SD) calculated, and unpaired (student) t-tests were performed to compare experimental groups with the corresponding wild-type control group. In cases where there were several percentage categories of phenotypes, the statistical comparison was performed between like categories, e.g. percentage increased were compared between control and each experimental treatment. In addition, ANOVA tests were performed to assess statistical significance in the context of comparing multiple samples.

## Acknowledgements

NIH Grant R01DK100237 to RAW supported this work. BED was also supported by a National Science Foundation Graduate Research Fellowship DGE-1313583. We are grateful to Elizabeth and Michael Gallagher for a generous gift to the University of Notre Dame on behalf of their family for the support of stem cell research. The funders had no role in the study design, data collection and analysis, decision to publish, or manuscript preparation. We thank the staffs of the Department of Biological Sciences and the Center for Zebrafish Research at the University of Notre Dame for their dedication and care of our zebrafish aquarium. Finally, we thank the members of our lab for their support, discussions, and insights about this work.

## Additional information

### Funding

| Funder | Grant reference number | Author |
|---|---|---|
| National Science Foundation | Graduate Research Fellowship - DGE-1313583 | Bridgette E Drummond |

| National Institutes of Health | R01DK100237 | Rebecca A Wingert |

The funders had no role in study design, data collection and interpretation, or the decision to submit the work for publication.

### Author contributions

SJP, CNC, RAW, Conception and design, Acquisition of data, Analysis and interpretation of data, Drafting or revising the article; JMC, BED, Acquisition of data, Analysis and interpretation of data

### Author ORCIDs

Rebecca A Wingert, http://orcid.org/0000-0003-3133-7549

### Ethics

Animal experimentation: Zebrafish were cared for and maintained in the Center for Zebrafish Research at the University of Notre Dame using experiments approved under protocol 16-025.

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
