## [Decision Letter]

Thank you for submitting your article "Prostaglandin signaling regulates nephron segment patterning of renal progenitors during vertebrate kidney development" for consideration by *eLife*. Your article has been reviewed by three peer reviewers, and the evaluation has been overseen by a Reviewing Editor (Tanya Whitfield) and Sean Morrison as the Senior Editor. The following individuals involved in review of your submission have agreed to reveal their identity: Dirk Meyer (Reviewer #1); Lilianna Solnica-Krezel (Reviewer #3).

The reviewers have discussed the reviews with one another and the Reviewing Editor has drafted this decision to help you prepare a revised submission.

Summary:

This is an interesting study on that addresses the factors that influence cell fate decisions in the developing vertebrate kidney, using the embryonic zebrafish pronephros as a model system. The authors have used a chemical genetic approach to identify prostaglandins as mediators that influence cell fate choice during development of the proximal and distal nephron segments. Results from the pharmacological screen have been validated through the use of morpholino-based gene knockdown to inhibit prostaglandin synthesis. The authors propose that the prostaglandin pathway acts upstream of the previously-identified transcriptional regulators Irx3b and Sim1a in the developing pronephros. The reviewers and editors agreed that the work was interesting and potentially significant. However, they also had some concerns that should be addressed to strengthen the manuscript, as outlined below.

Essential revisions:

1) Title: as suggested by reviewer 1, please change the word 'vertebrate' to 'zebrafish'. (Please note, however, that it is not essential to extend the work to other vertebrate species or different stages of kidney development.)

2) Given the known interactions between prostaglandin and other pathways such as Wnt signalling, there are concerns over the relevance of the phenotypes resulting from long-term treatment (4-22 hpf) with PG agonists for a nephron-specific patterning role of prostaglandins. These concerns are exacerbated by the rather mild nephric phenotypes described for prostaglandin pathway morphants. The option of indirect phenotypes therefore needs further discussion. In addition, experiments with later onset of agonist treatment are also required to exclude interference with general early embryonic patterning, for example by Wnt signalling. See comments from reviewer 1.

3) Following discussion between the reviewers, it was felt that a full analysis of any connection with Wnt signalling was not essential, but might be an interesting extension of the work in the future.

4) Measurements. There were concerns over the measurements and quantitation of the data, in particular, in the way that the extent of expression domains was measured (by a horizontal bar in terms of somite length, vs the actual length of the curved domain of gene expression shown). It was felt that more accurate measurements should be provided. In addition, n numbers should always accompany percentage values, wherever shown.

5) Effects on surrounding tissues. Please provide some assessment of the extent to which surrounding tissues are affected by the treatments shown.

6) The text needs tightening up in places, especially with reference to the distinction between PGs in general and PGE2 (see comments from reviewer 2).

7) For the experiments using morpholinos, it is important that these were done as rigorously as possible. Further supporting evidence is needed here (see comments from reviewer 3).

*Reviewer #1:*

In their manuscript Poureetezadi et al. use the zebrafish pronephros as a model to study functions of the prostaglandin pathway on nephron formation and patterning. Prostaglandins and in particular PGE2 have well documented functions in postnatal mammalian kidney development and renal physiology. In contrast, very little is known about activities in early kidney formation, mainly as genetic studies in mouse provided no major evidence for such requirements. In a screen for small compounds affecting zebrafish pronephron formation Poureetezadi et al. identified several prostaglandin agonists as agents causing caudally shifted or expanded tubule segments upon treatment of embryos between 4-24hpf. Detailed pharmacological analyses confirmed dose-dependent activities of PGE2 and the PGE2-derived PGB2 in tubule segmentation, and provided hints for a requirement of prostaglandin synthesizing enzymes COX1/2 and the PGE2 receptors Ptger2a/4a in the precise positioning of caudal segments. The pharmacological loss-of-function data were further confirmed by knockdown studies using morpholinos that supposedly prevent expression of COX1/2 and Ptger2a/4a proteins. Further, the authors show that nephric phenotypes seen after early embryonic interference with prostaglandin signaling correlate with changed expression of key transcriptional regulators of tubule segmentation and they provided evidence for a genetic role of prostaglandin signaling upstream of these factors. This interesting work is the first one demonstrating prostaglandin activities in nephric tubule patterning.

Unfortunately, the authors had addressed neither the underlying molecular mechanism nor the relevance for other vertebrate model systems or the metanephric kidney.

As pointed out by the authors, prostaglandins only recently became recognized as developmentally import fate determinants. In this context interference with canonical wnt-signaling had been identified as one of the most critical prostaglandin activities. Considering the importance of Wnt-signaling in kidney formation and patterning it is surprising that this connection had not been further analyzed.

The word 'vertebrate' in the title is not justified as the zebrafish is the only vertebrate that had been analyzed. Slattery et al. (2016) recently suggested a role of COX2 not only in postnatal but also in embryonic metanephric kidney development. While these mouse data support a possibly conserved requirement for prostaglandins, it should be noted that the phenotypic analyses of COX2 mutants was restricted to the glomerulus and no tubule data were provided.

The authors suggest a direct connection between the proposed nephron specific expression of ptger2a/4a (Figure 4—figure supplement.4) and the nephric loss and gain of function phenotypes. Expression of prostaglandin pathways components is not well documented and the experimental settings leave space for various alternative explanations. The images shown in Figure 4—figure supplement.4 suggest a restricted expression of ptger2a/4a in proximal tubule of 12-24ss embryos. In case of a direct function, pharmacological treatments starting at 12ss rather than at 4hpf should be sufficient to induce relevant gain and loss of function phenotypes. Additional data should be provided to confirm nephron specificity and to give details on the posterior-anterior extent of ptger2a/4a expression (for example sections and co-stains with nephric markers). Further, studies should be performed to determine the critical time-window of prostaglandin responsiveness.

*Reviewer #2:*

In the manuscript entitled "Prostaglandin signaling regulates nephron segment patterning of renal progenitors during vertebrate kidney development" by Poureetezadi et al., stemming from a chemical screening approach, the authors uncover a role for prostaglandin activity which impacts distal vs. proximal segment development of the pronephric tubules in the kidney. Specifically they found that exposure to prostaglandin agonists expanded the proximal segment (slc20a1a+) lineage and inhibited distal fate (slc12a1a+). In contrast inhibition of PG synthesis altered the distribution of specific distal fates. The authors also identified downstream consequences on key segment associated genes (irx3b and sim1a) that appeared to be influenced, directly or indirectly by PG modulation. The data presented throughout the paper strongly support the authors claims, however, descriptions of prostaglandin synthesis and signaling cascades are over-simplified, and general claims are made for "prostaglandins" when only PGE2-specific components are thoroughly investigated. As several types of prostaglandins were isolated in the screen, which can act in methods different from that investigated here (thromboxane receptor and PPARg), more precise language and a rationale for selecting PGE2 (not hit in the screen) would aid the reader. Similarly, while the data is nicely presented and logically organized to support the authors conclusions, the paper would benefit from addition of quantitative evaluations to aid appreciation by those outside of the field.

1) The authors over-simplify prostaglandin synthesis and signaling pathways, applying terminology and biological function relevant to PGE2 under the broad headline of "prostaglandins". This occurs as early as Figure 1, where they identify PGD2, PGA2, PGB2, and PGJ2 in the screen, which are known to stimulate thromboxane receptors (PGA2, B2) and PPARg (PGJ2), yet follow PGE2 (not a hit) and its relevant machinery. Similarly, Ptges is the enzyme for prostaglandin E2 synthesis (hence the "e" in its name, not general secondary processing), and Ptgers are the receptors for PGE2, not all of the PGs. While PTGS inhibition (Cox enzymes), could indeed broadly impact PG synthesis, the rest is quite specific to PGE2 and should not be generalized in the text, particularly when the screen hit several other potential modifiers as relevant to kidney biogenesis. This is not to say the data for PGE2 modification is wrong, it is just over generalized to "prostaglandins" and as such could cause issues for future investigations.

2) Given the independent signaling cascades associated with PGB2 and PGE2, it is surprising that exposure to both enzymes elicited the same biological effect. While receptor modulation is examined for 2 of the 4 PGE2 receptors, similar studies should be done in the context of exogenous PGE2 and PGB2 addition, as well as with the thromboxane receptors to confirm that addition of exogenous levels of each prostanoid doesn't cause errant signaling.

3) The doses utilized for dmPGE2 and Indomethacin are substantially higher than that found in other zebrafish or mouse papers, and well above physiological concentrations in humans. While the embryos appear grossly normal in whole mount images, it is important to document that alterations in kidney associated expression patterns are not simply due to gross morphological development issues or off target effects. Assessment of vascular markers by in situ or with a reporter line (Grosser et al) should be a quick way to test toxicity; receptor blocking analysis should confirm specificity.

4) While the overall conclusions seem well supported by the data shown, to make the paper more accessible to a general audience, the authors should use a more precise way to quantify the alterations in expression observed, or at least confirm the relevant ones, using ImageJ and/or a reporter line. The assays are currently very observational (with straight bars correlated with expression drawn on curvy embryos) and it is unclear how definitive statements like "50% reduction" are made when no quantification is shown.

*Reviewer #3:*

Starting with a chemical screen in developing zebrafish embryos, the Wingert and collaborators, identify components of prostaglandin signaling pathway as regulators of proximo-distal patterning of developing kidney. Using treatments with the pathway agonists (dmPGE2, PGB2) or inhibitors, and antisense morpholino oligonucleotides to impair expression of the PG synthetizing enzymes and their receptors (Ptger2a, Ptger4a), the authors provide evidence that prostaglandin signaling limits distal segment formation while promoting the proximal pronephric fates, acting upstream of Irx3b and Sim1a transcription factors, which have been previously implicated in this process. This is a very interesting work that expands our appreciation of the roles of prostaglandin signaling in cell fate specification during vertebrate embryogenesis. These roles were difficult to discern in the murine model, but recent studies in mouse and human point to PG involvement in kidney development and function. The authors leverage the experimental strengths of the zebrafish model, in which prostaglandin signaling can be easily manipulated during kidney development and the resulting effects can be monitored.

Whereas the proposed conclusions are significant and would be of interest to the developmental biology and renal research communities, they are not sufficiently supported by the presented data. Addressing the following questions and concerns would significantly strengthen the manuscript and make it suitable for publication.

One of the major concerns about the current manuscript is that there are many quantifications presented but without sufficient experimental detail to evaluate their statistical or biological significance. For several experiments (e.g. treatment of antagonists), there is insufficient experimental detail to evaluate the results and their interpretation. The quantifications of the phenotypes (reduction or expansion) of segments in various experimental regimens is given throughout the manuscript as% of embryos with altered expression (expanded or reduced). This shows that the fraction of affected embryos increases in a dose dependent manner but does not address whether the expressivity (degree of reduction or expansion) of the phenotype is also dose dependent. This is important given the overall conclusion that prostaglandins regulate segment patterning.

Were the measurements carried in a blinded fashion?

The number of embryos in individual experiments showing% of phenotype should be provided.

For Indomethacin treated embryos, the authors conclude that they "developed a 50% larger DE segment domain and a 20% smaller DL segment domain" referring to Figure 3 and its supplement. It is not clear where the data are presented on which this conclusion has been reached given that Figure 3 shows fractions of affected embryos and not dimensions of pronephric segments in control and treated embryos. Such conclusions need to be supported by clearly presented morphometric data, with the method of measurement clearly described, as well as the numbers of analyzed embryos.

The authors conclude that the changes in the dimensions of the pronephros proximo-distal segments represent the specific effects of prostaglandin signaling. However, the possibility of broader effects of prostaglandin signaling on embryonic dimensions and thus indirectly on pronephros is not sufficiently addressed. In the Methods the authors state "Gene domains were measured with respect to somite boundaries to assess pattern formation". However, the somitic staining that should provide such a reference is not clear in most of the figures. Given that earlier reports implicated prostaglandin signaling in morphogenetic movements of gastrulation (Gasser et al., PNAS, 2002; Cha et al., Genes & Dev, 2006; Speirs et al., Development, 2011), this needs to be carefully analyzed. Anteroposterior dimensions of the treated embryos should be measured as well as the relative size of pronephric segments to the neighboring somites for a reviewer/reader to evaluate.

In experiments in which PG signaling was inhibited with various agents, it is not clear at what developmental stages was this done? Given the previous work implicating PG signaling in gastrulation movements the timing of the treatment is important. It would be also of interest to identify when during embryogenesis PG signaling is required for segment patterning and when it can affect it.

Injection into 1 cell stage embryos of translation and splicing morpholinos targeting ptgs1 (cox1) and ptgs2a (cox2) enzymes is reported to affect pronephros patterning. However, it has been previously reported that ptgs1 morpholino injections caused severe epiboly defects (Gasser et al., 2002). In this work the morpholino experiments are insufficiently described. There is no information on the dosage of morpholinos per embryos (only stock concentration is given in methods), no dose response, no experiments describing the degree of loss of function. Yet, the authors conclude "these data indicate that endogenous levels of prostaglandins are critical for specification of the DE and DL segments". This conclusion is given before dmPGE2 rescue experiments are described.

Similar lack of detail and controls is a concern for the experiments in which two genes encoding the prostaglandin receptors, ptger2a and ptger4a are targeted using morpholinos. There are no data on the effectiveness of these reagents to downregulate expression of these genes and encoded proteins, or to lower the levels of prostaglandins. Since in recent years, significant concerns have been raised about the specificity of morpholinos (e.g. Kok et al., Dev Cell, 2015), the sparse details and controls for the experiments presented raise concerns.

In epistasis experiments presented in Figure 6 it would be important to see the effect of SC-580 alone. From the perspective of the model of PG – irx3b pathway proposed in Figure 7, the effect of dmPGE2 treatment that affect PST and DL but do not affect DE are confusing.

[Editors' note: further revisions were requested prior to acceptance, as described below.]

Thank you for resubmitting your work entitled "Prostaglandin signaling regulates nephron segment patterning of renal progenitors during zebrafish kidney development" for further consideration at *eLife*. Your revised article has been favorably evaluated by Sean Morrison (Senior editor) and Tanya Whitfield (Reviewing editor).

The manuscript has been substantially improved and all reviewers' comments have been addressed well. The additional experimental data help to strengthen and support the findings, and the writing is now clear and precise. Overall, this is a very interesting study and should be of wide interest.

One concern remains: the quantification has been improved but there was very little information about the statistical tests used for analysis of the data. t-tests are mentioned in the Materials and methods section, but these will not be suitable for analysis of some of the datasets, where there are multiple conditions and comparisons. Here, a test such as ANOVA, with appropriate post-test correction for multiple samples, would be preferred. Alternatively, the authors should clarify and justify the tests they have used. The tests that are used should be stated in each of the figure legends where appropriate, in addition to the statement in the Materials and methods.

---

## [Author Response]

*Essential revisions:*

*1) Title: as suggested by reviewer 1, please change the word 'vertebrate' to 'zebrafish'. (Please note, however, that it is not essential to extend the work to other vertebrate species or different stages of kidney development.)*

The authors agree and the recommended title change has been made.

*2) Given the known interactions between prostaglandin and other pathways such as Wnt signalling, there are concerns over the relevance of the phenotypes resulting from long-term treatment (4-22 hpf) with PG agonists for a nephron-specific patterning role of prostaglandins. These concerns are exacerbated by the rather mild nephric phenotypes described for prostaglandin pathway morphants. The option of indirect phenotypes therefore needs further discussion. In addition, experiments with later onset of agonist treatment are also required to exclude interference with general early embryonic patterning, for example by Wnt signalling. See comments from reviewer 1.*

We performed additional studies to address the time window when changes in PG levels affect nephron segment development. First, we treated embryos with either the antagonist indomethacin or agonist dmPGE_2_ starting at 12 ss with exposure lasting through to the 24 hpf time point. This indomethacin treatment elicited an expansion of the DE domain and reduction of the DL similar to that seen after the treatment lasting from 4 hpf to 24 hpf (new Figure 5). dmPGE_2_ treatment between 12 ss and 24 hpf induced an expansion of the PST and a DL reduction similar to the long dmPGE_2_ exposure time (new Figure 5). To quantify and further analyze these changes, we determined the absolute segment lengths and found the phenotypes following each PG alteration to be significant compared to controls, while the segment changes were indistinguishable from the corresponding drug exposure over the longer interval (new Figure 5). Further, we found that treatment with Ptger2 antagonists from the 12 ss to 24 hpf was also sufficient to induce a statistically significant DE expansion and DL reduction (new Figure 5—figure supplement 1), similar to the 4-24 hpf exposure (Figure 4—figure supplement 6; new Figure 4—figure supplement 9).

*3) Following discussion between the reviewers, it was felt that a full analysis of any connection with Wnt signalling was not essential, but might be an interesting extension of the work in the future.*

We have not pursued the connection to Wnt signaling in the current revision but agree that it will be an interesting extension for our future research.

*4) Measurements. There were concerns over the measurements and quantitation of the data, in particular, in the way that the extent of expression domains was measured (by a horizontal bar in terms of somite length, vs the actual length of the curved domain of gene expression shown). It was felt that more accurate measurements should be provided. In addition, n numbers should always accompany percentage values, wherever shown.*

We now provide absolute length measurements for all nephron segment phenotypes (Please see Figure 2; Figure 2—figure supplement 2; Figure 3; Figure 3—figure supplement 1; Figure 4—figure supplement 9). Data involving phenotype penetrance and dosage responses have been moved to supplemental figures. We provide n values in either the graph or legend for all quantitations.

*5) Effects on surrounding tissues. Please provide some assessment of the extent to which surrounding tissues are affected by the treatments shown.*

We assessed surrounding tissues following PG agonist (dmPGE_2,_ 100 µM) and PG antagonist (indomethacin, 30 µM) exposure by whole mount in situ hybridization (as suggested by reviewer #2), and also examined circulation within the vasculature. We did not observe differences *flk1* or *gata1* transcript expression at 24 hpf in embryos exposed to DMSO, dmPGE_2_ or indomethacin between 4 hpf and 24 hpf, indicating normal vascular and primitive blood development, respectively (new Figure 2—figure supplement 4). This is consistent with a published study that reported effects of indomethacin at 40 µM, but normal development of the vasculature following indomethacin treatment at 25 µM (Cha, et al., 2005).

Next, we performed o-dianisidine (benzidene) staining in embryos treated with DMSO, dmPGE_2_ or indomethacin between 4 hpf and 24 hpf, which labels hemoglobinized erythrocytes and provides a sensitive assessment of defects in circulation or vascular integrity that can be undetected when using only live imaging with stereomicroscopy. Blood flow in these drug treated embryos was equivalent to wild-type control embryos through the 48-55 hpf stage, and we did not observe compromised vessel integrity or hematomas (e.g. bleeding, blood pooling) with o-dianisidine staining (new Figure 2—figure supplement 4).

*6) The text needs tightening up in places, especially with reference to the distinction between PGs in general and PGE2 (see comments from reviewer 2).*

We have revised the manuscript extensively to address this weakness and to be more specific throughout.

*7) For the experiments using morpholinos, it is important that these were done as rigorously as possible. Further supporting evidence is needed here (see comments from reviewer 3).*

In the revision, we provide the data that morpholinos used to interfere with mRNA splicing of *ptgs1, ptgs2a, ptger2a* and *ptger4a* caused retention of intronic sequences that disrupt the coding sequence of each gene (new Figure 4—figure supplement 10). A full description can be found in the corresponding text and figure legend.

*Reviewer #1:*

*[…] As pointed out by the authors, prostaglandins only recently became recognized as developmentally import fate determinants. In this context interference with canonical wnt-signaling had been identified as one of the most critical prostaglandin activities. Considering the importance of Wnt-signaling in kidney formation and patterning it is surprising that this connection had not been further analyzed.*

We agree that it would be interesting to delineate the relationship between prostaglandin signaling and canonical Wnt-signaling in relation to kidney development. However, we agree with the collective decision of the reviewers that this interrogation is beyond the scope of our present study.

*The word 'vertebrate' in the title is not justified as the zebrafish is the only vertebrate that had been analyzed. Slattery et al. (2016) recently suggested a role of COX2 not only in postnatal but also in embryonic metanephric kidney development. While these mouse data support a possibly conserved requirement for prostaglandins, it should be noted that the phenotypic analyses of COX2 mutants was restricted to the glomerulus and no tubule data were provided.*

We have changed the title to "Prostaglandin signaling regulates nephron segment patterning of renal progenitors during zebrafish kidney development" to more specifically describe the context of the study.

*The authors suggest a direct connection between the proposed nephron specific expression of ptger2a/4a (Figure 4—figure supplement 4) and the nephric loss and gain of function phenotypes. Expression of prostaglandin pathways components is not well documented and the experimental settings leave space for various alternative explanations. The images shown in Figure 4—figure supplement 4 suggest a restricted expression of ptger2a/4a in proximal tubule of 12-24ss embryos. In case of a direct function, pharmacological treatments starting at 12ss rather than at 4hpf should be sufficient to induce relevant gain and loss of function phenotypes.*

Pharmacological treatments have been performed between the 12 ss to 24 hpf for dmPgE2, indomethacin, and the Ptger2 antagonists. Treatment with either indomethacin or the Ptger antagonists between 12 ss and 24 hpf was sufficient to cause an expanded DE and reduced DL, as seen in treatments from 4 hpf to 24 hpf. dmPgE2 treatment between 12 ss and 24 hpf cause an equivalent expansion of the PST and a restriction of the DL as treatment between the 4 hpf to 24 hpf time. These findings are consistent with the notion that Ptger2a/4a act in zebrafish embryo renal progenitors between the 12 ss and 24 hpf to influence segment patterning. Descriptions of these studies have been added to the revised Results section, and the new data are located in new Figure 5 and new Figure 5—figure supplement 1.

*Additional data should be provided to confirm nephron specificity and to give details on the posterior-anterior extent of ptger2a/4a expression (for example sections and co-stains with nephric markers). Further, studies should be performed to determine the critical time-window of prostaglandin responsiveness.*

We revised Figure 4—figure supplement 4 to provide details of the anterior-posterior domain of *ptger2a* and *ptger4a* expression in the intermediate mesoderm with respect to the somites. Transcripts for each gene are initially detected in putative renal progenitors situated adjacent to somites 5 through 10 at the 12 ss stage, and as somitogenesis progresses they are both detected in a domain situated adjacent to somites 5 through 13 until approximately the 24 ss.

We did perform exhaustive testing of various *ptger2a/4a* probes for fluorescent WISH detection but were not able to determine the appropriate conditions for successful fluorescent labeling, which would have facilitated the most precise co-localization studies and confocal imaging. However, using standard WISH we were able to validate that *ptger2a* and *ptger4a* transcripts are expressed in the stripes of pronephros progenitors within the intermediate mesoderm due to their colocalization with *cadherin-17 (cdh17)* transcripts (Figure 4—figure supplement 4).

*Reviewer #2:*

*[…] 1) The authors over-simplify prostaglandin synthesis and signaling pathways, applying terminology and biological function relevant to PGE2 under the broad headline of "prostaglandins". This occurs as early as Figure 1, where they identify PGD2, PGA2, PGB2, and PGJ2 in the screen, which are known to stimulate thromboxane receptors (PGA2, B2) and PPARg (PGJ2), yet follow PGE2 (not a hit) and its relevant machinery. Similarly, Ptges is the enzyme for prostaglandin E2 synthesis (hence the "e" in its name, not general secondary processing), and Ptgers are the receptors for PGE2, not all of the PGs. While PTGS inhibition (Cox enzymes), could indeed broadly impact PG synthesis, the rest is quite specific to PGE2 and should not be generalized in the text, particularly when the screen hit several other potential modifiers as relevant to kidney biogenesis. This is not to say the data for PGE2 modification is wrong, it is just over generalized to "prostaglandins" and as such could cause issues for future investigations.*

We provided the simplified prostaglandin synthesis and signaling pathway in Figure 1, focusing on PGE_2_ relevant molecules, as an example to help orient readers. The accompanying text for this section was partly incorrect due to poor editing, and has now been revised for accuracy.

In the revision, we have endeavored to be more specific throughout. For example, the Abstract has been modified to articulate more precisely the manipulations and phenotypes. The Introduction has been edited to provide a better description of prostanoid production and to clearly designate that Figure 1 is an example of PGE_2_. Throughout the results, we revised our language to be specific when discussing PGE_2_, particular enzymes and receptors.

*2) Given the independent signaling cascades associated with PGB2 and PGE2, it is surprising that exposure to both enzymes elicited the same biological effect.*

We agree. However, it has been shown that different prostaglandins after certain thresholds of concentrations will freely interact with other prostaglandin receptors (Kiriyama et al., 1997, Tootle, 2013). This may explain why other prostaglandin compounds, such as PGB_2_, were capable of eliciting a similar effect as PGE_2_. One interpretation of the dosage-sensitive segment phenotypes is that they are related to this phenomenon. However, further studies are needed to truly distinguish specific effects of particular prostaglandins, such as by testing loss of function of particular prostaglandin synthases and the appropriate prostaglandin receptors during pronephros development.

*While receptor modulation is examined for 2 of the 4 PGE2 receptors, similar studies should be done in the context of exogenous PGE2 and PGB2 addition, as well as with the thromboxane receptors to confirm that addition of exogenous levels of each prostanoid doesn't cause errant signaling.*

We performed the experiments to test how addition of PGE_2_ affected the nephron phenotype in ptger2a and ptger4a deficient zebrafish (new Figure 4—figure supplement 8). Addition of PGE_2_ did not alter either the phenotype of ptger2a or ptger4a knockdowns, where we observed similar DE expansions and DL reductions as DMSO treated controls (new Figure 4—figure supplement 8). This was confirmed by absolute segment length measurements and statistical analysis (new Figure 4—figure supplement 8). While examination of thromboxane receptors would be another interesting area to explore, we believe it is beyond the scope of the present work.

*3) The doses utilized for dmPGE2 and Indomethacin are substantially higher than that found in other zebrafish or mouse papers, and well above physiological concentrations in humans. While the embryos appear grossly normal in whole mount images, it is important to document that alterations in kidney associated expression patterns are not simply due to gross morphological development issues or off target effects. Assessment of vascular markers by in situ or with a reporter line (Grosser et al) should be a quick way to test toxicity; receptor blocking analysis should confirm specificity.*

We assessed surrounding tissues following PG agonist (dmPGE_2,_ 100 µM) and PG antagonist (indomethacin, 30 µM) exposure by whole mount in situ hybridization, and also examined vascular integrity at later stages. We did not observe differences *flk1* or *gata1* transcript expression at 24 hpf in embryos exposed to DMSO, dmPGE_2_ or indomethacin between 4 hpf and 24 hpf, indicating normal vascular and primitive blood development, respectively (new Figure 2—figure supplement 4). This is consistent with a published study that reported effects of indomethacin at 40 µM, but normal development of the vasculature following indomethacin treatment at 25 µM (Cha, et al., 2005). We also performed o-dianisidine (benzidene) staining in embryos treated with DMSO, dmPGE_2_ or indomethacin between 4 hpf and 24 hpf, which labels hemoglobinized erythrocytes and provides a sensitive assessment of defects in circulation or vascular integrity that can be undetected when using only live imaging with stereomicroscopy. With o-dianisidine staining, we observed that the drug treated embryos had equivalent vascular integrity compared to wild-type control embryos through the 48-55 hpf stage, such that we did not observe blockages or hematomas (e.g. bleeding, blood pooling) (new Figure 2—figure supplement 4).

*4) While the overall conclusions seem well supported by the data shown, to make the paper more accessible to a general audience, the authors should use a more precise way to quantify the alterations in expression observed, or at least confirm the relevant ones, using ImageJ and/or a reporter line. The assays are currently very observational (with straight bars correlated with expression drawn on curvy embryos) and it is unclear how definitive statements like "50% reduction" are made when no quantification is shown.*

To make the paper more accessible, we now provide absolute segment length values in addition to our evaluation of segment domains relative to the somite number. We measured segment lengths in microns for the pertinent segments that change in drug treatments, morphants, and mutants using Nikon Software.

*Reviewer #3:*

*[…]Whereas the proposed conclusions are significant and would be of interest to the developmental biology and renal research communities, they are not sufficiently supported by the presented data. Addressing the following questions and concerns would significantly strengthen the manuscript and make it suitable for publication.*

*One of the major concerns about the current manuscript is that there are many quantifications presented but without sufficient experimental detail to evaluate their statistical or biological significance. For several experiments (e.g. treatment of antagonists), there is insufficient experimental detail to evaluate the results and their interpretation. The quantifications of the phenotypes (reduction or expansion) of segments in various experimental regimens is given throughout the manuscript as% of embryos with altered expression (expanded or reduced). This shows that the fraction of affected embryos increases in a dose dependent manner but does not address whether the expressivity (degree of reduction or expansion) of the phenotype is also dose dependent. This is important given the overall conclusion that prostaglandins regulate segment patterning.*

Within Figure 2—figure supplement 2 – different colored bars represent the degree of expansion or reduction as indicated by the legend and suggest that indeed, as prostaglandin concentration is increased there is an expansion of the PST up to 6 somites in domain length from that of the WT state of 3 corresponding somites. Additionally, we performed measurements of domain lengths of segments throughout the revised manuscript, which we believe provides the most clear description of the segment phenotypes.

*Were the measurements carried in a blinded fashion?*

*The number of embryos in individual experiments showing% of phenotype should be provided.*

All experiments showing% phenotype consisted of a minimum of at least 15 embryos per group in triplicate, which we have notated in the figure legends for the most complex data sets.

*For Indomethacin treated embryos, the authors conclude that they "developed a 50% larger DE segment domain and a 20% smaller DL segment domain" referring to Figure 3 and its supplement. It is not clear where the data are presented on which this conclusion has been reached given that Figure 3 shows fractions of affected embryos and not dimensions of pronephric segments in control and treated embryos. Such conclusions need to be supported by clearly presented morphometric data, with the method of measurement clearly described, as well as the numbers of analyzed embryos.*

In the revision we now provide absolute length measurements for all nephron segment phenotypes (Please see Figure 2; Figure 2—figure supplement 2; Figure 3; Figure 3—figure supplement 1; Figure 4—figure supplement 9). Data involving phenotype penetrance and dosage responses have been moved to supplemental figures. We provide n values in either the graph or legend for all quantitations.

*The authors conclude that the changes in the dimensions of the pronephros proximo-distal segments represent the specific effects of prostaglandin signaling. However, the possibility of broader effects of prostaglandin signaling on embryonic dimensions and thus indirectly on pronephros is not sufficiently addressed. In the Methods the authors state "Gene domains were measured with respect to somite boundaries to assess pattern formation". However, the somitic staining that should provide such a reference is not clear in most of the figures. Given that earlier reports implicated prostaglandin signaling in morphogenetic movements of gastrulation (Gasser et al., PNAS, 2002; Cha et al., Genes & Dev, 2006; Speirs et al., Development, 2011), this needs to be carefully analyzed. Anteroposterior dimensions of the treated embryos should be measured as well as the relative size of pronephric segments to the neighboring somites for a reviewer/reader to evaluate.*

At the dosages for chemical treatments and various knockdowns, we did not observe noticeable morphological changes. To actually document this and provide quantification as well, we performed measurements to acquire absolute values in microns using Nikon Software for the pronephric domain (where the PCT begins and the DL ends), as well as for the anterior posterior axis of the embryos which we have denoted as "tip to tail" for dmPgE2, indomethacin, and 1% DMSO treated embryos. In both cases, we found no statistical differences between the groups (new Figure 2—figure supplement 3). These data support the conclusion that the segment domains changes are not simply a product of embryo dimension changes, but rather a modulation of segment specific identities.

*In experiments in which PG signaling was inhibited with various agents, it is not clear at what developmental stages was this done? Given the previous work implicating PG signaling in gastrulation movements the timing of the treatment is important. It would be also of interest to identify when during embryogenesis PG signaling is required for segment patterning and when it can affect it.*

We have revised the manuscript to ensure that the time window for chemical treatments was made more apparent in the results rather than having this stated in the methods. Next, we agreed that the perturbation of the prostaglandin pathway at later time windows was critical to assess. To address this, we treated embryos with dmPgE2, indomethacin, and 1% DMSO between the 12 ss to 24 hpf. In all cases, we observed that these treatments phenocopied their corresponding treatments from 4 hpf to 24 hpf (new Figure 5). Additionally, we treated embryos with the ptger2a antagonists (AH6809 and PF04418948) from the 12ss to 24hpf and saw a similar increase of the DE and decrease of the DL as seen in treatments from 4hpf to 24hpf (new Figure 5—figure supplement 1).

*Injection into 1 cell stage embryos of translation and splicing morpholinos targeting ptgs1 (cox1) and ptgs2a (cox2) enzymes is reported to affect pronephros patterning. However, it has been previously reported that ptgs1 morpholino injections caused severe epiboly defects (Gasser et al., 2002). In this work the morpholino experiments are insufficiently described. There is no information on the dosage of morpholinos per embryos (only stock concentration is given in methods), no dose response, no experiments describing the degree of loss of function. Yet, the authors conclude "these data indicate that endogenous levels of prostaglandins are critical for specification of the DE and DL segments". This conclusion is given before dmPGE2 rescue experiments are described.*

*Similar lack of detail and controls is a concern for the experiments in which two genes encoding the prostaglandin receptors, ptger2a and ptger4a are targeted using morpholinos. There are no data on the effectiveness of these reagents to downregulate expression of these genes and encoded proteins, or to lower the levels of prostaglandins. Since in recent years, significant concerns have been raised about the specificity of morpholinos (e.g. Kok et al., Dev Cell, 2015), the sparse details and controls for the experiments presented raise concerns.*

We have modified our Methods and figures/figure legends to more clearly describe our morpholino related experiments. While most of morpholinos described in the present work have been used in a number of other published studies (Grosser et al., 2002, Cha et al., 2006, North et al., 2008, Nissim et al., 2014), we do recognize the need for morpholino work to be verified. Indeed, part of our rationale for extensive parallel studies with chemical genetics and use of multiple knockdown reagents was to study these factors with independent tools. Unfortunately, we are unable at the present time to assess the consequential protein levels of the genes targeted by morpholinos that interact with 5’UTR or translational start site due to the lack of appropriate specific antibodies. However, in the revision, we provide a thorough transcriptional analysis to document the effects of splice morpholinos used for knockdown of *ptgs1, ptgs2a, ptger2a,* and *ptger4a* using RT-PCR (new Figure 4—figure supplement 10). In all cases, we were able to confirm that incorrect splicing occurred as a product of each splice morpholino or combination (new Figure 4—figure supplement 10). In fact, for each gene, we found that the morpholino microinjection eliminated the presence of wild-type transcripts. In the case of *ptger2a* and *ptger4a*, we needed to obtained splice targeting morpholinos and we have validated their phenotypic effects on the pronephros (new Figure 4—figure supplement 10).

*In epistasis experiments presented in Figure 6 it would be important to see the effect of SC-580 alone.*

We provide data on the effect of the selective Cox1 inhibitor in Figure 4, with the absolute segment length changes documented in Figure 4—figure supplement 9. In Figure 6 we provide a representative image to remind readers of the DE phenotype from exposure to this compound.

*From the perspective of the model of PG – irx3b pathway proposed in Figure 7, the effect of dmPGE2 treatment that affect PST and DL but do not affect DE are confusing.*

The working models depict the effects of PGE_2_ on segment size, where (A) PGE_2_ biosynthesis is required to restrict the DE by inhibiting the domain of *irx3b* expression and to promote the DL, and (B) PGE_2_ applied exogenous is capable of positively modulating the size PST segment by promoting *sim1a* expression, and meanwhile inhibiting the DL. In the case of (B), the DE segment develops at a normal size but the position is altered along the anterior-posterior axis, likely due to the PST expansion and the shifted domain of *irx3b* expression, and thus we thought that drawing an arrow between PGE_2_ and the DE in this case would be misleading compared to the other arrows and phenotypes because the DE segment size in this case is normal. We edited the figure legend to remind readers that we correlated a change in the *irx3b* domain to the shift in the DE location, and that changes in the *mecom* expression domain correlate with the DL reduction in both (A) and (B).

[Editors' note: further revisions were requested prior to acceptance, as described below.]

*[…] The manuscript has been substantially improved and all reviewers' comments have been addressed well. The additional experimental data help to strengthen and support the findings, and the writing is now clear and precise. Overall, this is a very interesting study and should be of wide interest.*

*One concern remains: the quantification has been improved but there was very little information about the statistical tests used for analysis of the data. t-tests are mentioned in the Materials and methods section, but these will not be suitable for analysis of some of the datasets, where there are multiple conditions and comparisons. Here, a test such as ANOVA, with appropriate post-test correction for multiple samples, would be preferred. Alternatively, the authors should clarify and justify the tests they have used. The tests that are used should be stated in each of the figure legends where appropriate, in addition to the statement in the Materials and methods.*

We have revised the Materials and methods as requested, and we have edited each figure legend to state which statistical analysis was used.

With regard to our statistical tests: In many figures, we utilized unpaired t-tests to compare nephron segment lengths between two samples. However, in cases of 3 or more samples, we now report the outcome of ANOVA tests rather than t-tests, as ANOVA is preferred for comparisons across multiple samples. We have now revised the following figures with ANOVA results: Figure 2—figure supplement 3; Figure 4—figure supplement 8; Figure 4—figure supplement 10; Figure 5; Figure 5—figure supplement 1; and Figure 6. These revisions provide a more appropriate statistical analysis for numerous aspects of the work.

In contrast, however, in the graphs reporting the percentage phenotype following drug or morpholino treatments, we focused on the key comparison of each individual phenotype to the corresponding wild-type control. For this, we utilized unpaired t-tests, and have revised the figure legends to clearly indicate our analysis. While it might be interesting to look across groups in some cases, our primary goal of these statistical analyses was to determine whether PGE_2_ treatment or alterations of endogenous prostaglandin biosynthesis significantly altered nephron segments compared to wild-type control embryos, as opposed to delving into the cross-treatment comparisons.